# Role of Glycolytic and Glutamine Metabolism Reprogramming on the Proliferation, Invasion, and Apoptosis Resistance through Modulation of Signaling Pathways in Glioblastoma

**DOI:** 10.3390/ijms242417633

**Published:** 2023-12-18

**Authors:** Cristina Trejo-Solis, Daniela Silva-Adaya, Norma Serrano-García, Roxana Magaña-Maldonado, Dolores Jimenez-Farfan, Elizabeth Ferreira-Guerrero, Arturo Cruz-Salgado, Rosa Angelica Castillo-Rodriguez

**Affiliations:** 1Laboratorio Experimental de Enfermedades Neurodegenerativas, Laboratorio de Reprogramación Celular, Departamento de Neurofisiología, Instituto Nacional de Neurología y Neurocirugía, Ciudad de Mexico 14269, Mexico; danieladaya@ciencias.unam.mx (D.S.-A.); norma.serrano@innn.edu.mx (N.S.-G.); rmagana@innn.edu.mx (R.M.-M.); 2Laboratorio de Inmunología, División de Estudios de Posgrado e Investigación, Facultad de Odontología, Universidad Nacional Autónoma de México, Ciudad de Mexico 04510, Mexico; farfanmd@unam.mx; 3Centro de Investigación Sobre Enfermedades Infecciosas, Instituto Nacional de Salud Pública, Cuernavaca 62100, Mexico; elizabeth.ferreira@insp.mx (E.F.-G.); tlacaelel333@yahoo.com (A.C.-S.); 4CICATA Unidad Morelos, Instituto Politécnico Nacional, Boulevard de la Tecnología, 1036 Z-1, P 2/2, Atlacholoaya 62790, Mexico

**Keywords:** glioma, metabolism, glucose, glutamine, oncogenic pathways

## Abstract

**Simple Summary:**

Metabolic reprogramming in glioblastoma is a hallmark of malignancy. Metabolic enzymes and metabolites participate in energetic metabolism and function as signaling molecules associated with genomic instability, mutations, epigenetic changes, and the activation of oncogenic signaling pathways. In addition, metabolites obtained from the metabolic network regulate several metabolic enzymes facilitating malignancy, therapeutic resistance, and tumor recurrence at the transcriptional and post-translational levels. In this review, we described the role of the glycolytic and glutamine metabolic network as an inductor of the cellular pathophysiology of glioblastoma as well as a therapeutic target in glioma cells.

**Abstract:**

Glioma cells exhibit genetic and metabolic alterations that affect the deregulation of several cellular signal transduction pathways, including those related to glucose metabolism. Moreover, oncogenic signaling pathways induce the expression of metabolic genes, increasing the metabolic enzyme activities and thus the critical biosynthetic pathways to generate nucleotides, amino acids, and fatty acids, which provide energy and metabolic intermediates that are essential to accomplish the biosynthetic needs of glioma cells. In this review, we aim to explore how dysregulated metabolic enzymes and their metabolites from primary metabolism pathways in glioblastoma (GBM) such as glycolysis and glutaminolysis modulate anabolic and catabolic metabolic pathways as well as pro-oncogenic signaling and contribute to the formation, survival, growth, and malignancy of glioma cells. Also, we discuss promising therapeutic strategies by targeting the key players in metabolic regulation. Therefore, the knowledge of metabolic reprogramming is necessary to fully understand the biology of malignant gliomas to improve patient survival significantly.

## 1. Introduction

Brain tumors are highly aggressive and lethal, with an incidence of 6–7 cases per 100,000 persons per year [1]. Gliomas are the main neoplasia that affects the central nervous system (CNS), accounting for approximately 50% of all brain neoplasms [2]. The World Health Organization classifies gliomas as oligodendrogliomas, oligoastrocytomas, and astrocytomas being the most prevalent according to their histological characteristics. Astrocytomas can be classified according to their degree of malignancy and the severity of their clinical signs into four grades: grade I (pilocytic), grade II (diffuse), grade III (anaplastic), and glioblastoma (GBM; grade IV) [2]. Grade I and II gliomas are low-grade gliomas, with good prognosis whereas Grade III and IV gliomas are high-grade gliomas with bad prognosis, glioblastoma being the most malignant [2].

Recently, the classification of GBM tumors has shifted toward molecular and genetically defined subtypes [3]. The glioblastomas are characterized by poorly differentiated neoplastic astrocytes with cellular polymorphism, nuclear atypia, high mitotic activity, hypoxia, vascular proliferation, thrombosis, and epigenetics, and genetic heterogeneity [2]. These tumors exhibit great local invasiveness, glial tumor infiltration at 1–2 cm from the original tumor mass prevents a total tumor resection after surgery and results in a high rate of tumor recurrence [4]. GBM remains refractory to surgical treatment, radiotherapy, systemic or local chemotherapy, and immunotherapy [5]. Epidemiological studies have reported that patient survival is approximately 12–15 months after diagnosis, despite the treatment [6]. 

The resistance of GBM to standard treatment is mainly due to a highly mutated genome led by mutations, amplifications, deletions, and translocations of genes, which promote the inactivation of suppressor of tumors (neurofibromin 1 (NF1), p53, p14^ARF^, retinoblastoma (RB), phosphatase, and tensin homolog deleted on chromosome 10 (PTEN), tuberous sclerosis complex 1 (TSC1), TSC2, parkin RBR E3 ubiquitin-protein ligase (PARK2), and cyclin-dependent kinase inhibitor 2A,-2B (CDKN2A,-2B) as well as the overactivation of tyrosine kinase receptors such as EGFR/vIII, c-MET, PDGR, insulin-like growth factor (IGF), and the vascular endothelial growth factor receptor (VEGFR), which are upregulated in GBM [7,8,9]. These receptors induce the activation of the Ras/RAF/mitogen-activated protein kinase (MEK)/extracellular-signal-regulated kinase (ERK), and phosphoinositide 3-kinase (PI3K)/protein kinase B (AKT)/mammalian target of rapamycin (mTOR) signaling pathways [8,9]. Moreover, these signaling pathways lead to the activation of transcriptional factors such as hypoxia-inducible factor-1 (HIF-1), and c- myc as well as the inactivation of p53 and PTEN to regulate cell proliferation, angiogenesis, invasion, apoptosis, autophagy, and metabolism. 

The metabolic profile has been used to subclassify gliomas finding relevance in the survival rate of the patients, revealing that metabolic reprogramming is a factor in the development of GBM [10]. Higher rate glycose consumption, lactate production in the presence of oxygen (Warburg effect), and addiction to glutamine are reported in glioma cells, which support the bioenergetic requirements as well as the formation of intermediate metabolite necessary for the biosynthetic demands of lipids, carbohydrates, protein, DNA and RNA and appropriate redox status of glioma cells higher proliferative [5,10]. Furthermore, the metabolites obtained during metabolic reprogramming can function as signaling molecules to allow crosstalk between metabolic pathways and among metabolic, epigenetic, and signaling pathways [11].

In this review, we summarized and discussed current knowledge about advances in the metabolism of glioblastoma cells on the expression and activity of canonic and non-canonical pathways of metabolic enzymes as well as of metabolic products, which participate in the regulation of the Warburg effect, proliferation, survival, epithelial–mesenchymal transition, migration, invasion, apoptosis, autophagy, epigenetic, and genic modulation and other cellular functions thereby promoting the initiation, promotion, and progression for glioma [12,13]. Understanding these metabolic alterations is crucial for enhancing the development of novel therapeutic strategies for glioblastoma. Furthermore, we described experimental drugs that block the metabolism in glioblastoma treatment.

## 2. Glycolysis in Cancer Cells

Glycolysis is not only related to the production of energy, but also the synthesis of intermediates of other anabolic pathways, and is consequently linked to cell proliferation [14] (Figure 1). Pyruvate could be converted into lactate by the lactate dehydrogenase (LDH), but it could also be converted into the amino acid alanine (Ala) through the alanine aminotransferase (AAT) or converted to acetyl-CoA by the pyruvate dehydrogenase (PDH) to participate in the tricyclic acid cycle (TCA) and the oxidative phosphorylation [15]. Alternatively, G6P could be redirected to the pentose phosphate pathway (PPP), mainly regulated by the glucose-6-phosphate dehydrogenase (G6PDH), to generate carbohydrates such as ribose-5-phosphate to the ribonucleotide synthesis and produce nicotinamide adenine dinucleotide phosphate (NADPH) coupled to the glutathione system to avoid the oxidative stress. Also, the G6P is a precursor for the synthesis of glycogen and the generation of UDP-glucuronate, which is necessary for glycoproteins and proteoglycan synthesis [16]. Glycolysis also provides glyceraldehyde-3-phosphate (GA3P) and dihydroxyacetone phosphate (DHAP) precursors for the synthesis of triglycerides and lipids [17]. Additionally, amino acids such as glycine (Gly), cysteine (Cys), and serine (Ser) are obtained from GA3P [17,18]. It has been demonstrated that 3PG binds to and inhibits 6-phosphogluconate dehydrogenase (6PGDH), an enzyme que participle in the pentose phosphate pathway [19]. 2PG activates 3-phosphoglycerate dehydrogenase (3PGDH) in the serine biosynthesis pathway [19]. 

A link between the glycolytic pathway and TCA has been established with citrate as an intermediary and documented previously [20,21,22]. Citrate is an inhibitor of PFK1, a key regulator of glycolysis, and PFK2, which produces F2,6-BP, an activator of PFK1. Then, the interaction between both pathways is well established. Moreover, citrate as metabolite is linked to the grade of aggressiveness in cancer through the apoptotic process. It has been reported that citrate inhibits the expression of Bcl-xL and Mcl-1, both members of the anti-apoptotic Bcl-2 family. Moreover, citrate induces the expression of pro-apoptotic proteins such as Bax, caspase 3, and 9 [20]. Then, the regulation of both pathways is a key factor in the resistance of apoptosis. Furthermore, glycolytic intermediates are precursors for the synthesis of acyl tricylglycerides, cholesterol, and fatty acids (Figure 1).

In addition, F1,6-BP increases the Warburg effect by inhibiting the activities of the III and IV complexes of the electron transport chain (ETC) [23] and activating the Ras/PI3K/AKT signaling pathway due to its direct binding to Son-of-Sevenless-homolog-1 (SOS1) and EGFR [24,25], both activators of Ras pathway [25,26]. Ras/PI3K/AKT signaling leads to the activation of the HIF-1 transcriptional factor, LDHA, and PFK2 [27]. PFK2 catalyzes the transformation of F6P to F2,6BP, which is an activator of PFK1 [28], supporting the formation of lactate in cancer cells [28]. The high amounts of lactic acid are expelled by the monocarboxylate transporters (MCTs) present in tumor cells, which acidify the tumor microenvironment inducing the death of normal cells surrounding the tumor and improving the proliferation, invasion, and migration of tumor cells [29,30]. In addition, the lactate is a metabolic fuel for tumoral cells which induces VEGF overexpression and supports the tumoral angiogenesis [31]. Lactate also protects the tumoral cells from the immunosurveillance of NK cells [32], nuclear polymorphs, and myeloid-derived suppressor cells (MDSCs) which are recruited to hypoxic areas of the tumor [32]. Moreover, lactate reduces the proliferation, production of cytokines, and cytotoxicity of T lymphocytes [33]. At the same time, lactate and pyruvate allow the stabilization of HIF-1, providing a potential feed-forward loop that may be critical to aerobic glycolysis [34]. In well-oxygenated environments, HIF-1 subunits are hydroxylated at conserved proline residues by prolyl hydroxylase domain-containing enzymes (PHDs). Hydroxylated HIF-1 is recognized and marked for proteasomal destruction by an E3 ubiquitin ligase, the von Hippel–Lindau protein (pVHL) complex. When the cell is under hypoxic stress, PHDs activity is diminished, and stabilized HIF-1 proteins can induce the transcription of genes with adaptive functions [35,36]. HIF-1 can activate anti-apoptotic and pro-proliferation genes, contributing to tumor formation of cancer cells [37,38]. On the other hand, glycolysis can be a very valuable source of ATP when cells are subjected to low oxygen tensions, as occurs when tumor enlargement and disorganized growth of new vessels (angiogenesis) produce large hypoxic areas [29] or when mitochondria already do not produce ATP due to the higher rate of mutations in its DNA and malfunction of the electron transport chain (ETC) [39]. Interestingly, metabolic modifications due to altered enzymes of the glycolysis and Warburg effect have been described extensively in glioblastomas, which are characterized by their aggressiveness and resistance to treatment [40]. In this review, we will focus on the new proposals with respect to this subject.

### 2.1. The Warburg Effect and Glucose Avidity in Glioblastoma

The Warburg effect has been described as a general characteristic of cancer. A recent study based on the transcriptomic profile of cancer samples confirmed a glycolytic pattern characterized by upregulation of glucose and amino acid transporters and increased glycolysis and pentose phosphate pathways, in contrast to the decrease of β-oxidation and amino acid synthetic pathways [41]. However, specific differences between tumors have also been reported, particularly in glioblastoma where approximately a three-fold increase in the glycolytic process compared to the normal brain has been observed [42]. In this regard, glioblastomas are avid of glucose uptake as has been demonstrated by the measurement of ^18^fluoro-2-deoxyglucose uptake (FDG) by positron emission tomography (PET) [43] as well as [U-^13^C] glucose by nuclear magnetic resonance (NMR) spectroscopy in glioma samples [44]. The analysis of [U-^13^C] glucose uptake also revealed that nearly 50% of the acetyl-CoA metabolites come from glucose; consequently, other substrates besides glucose should be involved in the energetic maintenance of the gliomas [44]. Recently, a Warburg index using imaging tools has been proposed. The Warburg index uses the lactate concentration obtained by proton magnetic resonance spectroscopy (^1^H-MRS) divided by glucose uptake by FDG-PET. A second option is the use of deuterium metabolic imaging (DMI). Both options could be used to characterize the Warburg effect in gliomas in patients and orientate the prognosis and treatment [45].

The mechanisms underlying glucose dependence in gliomas have also been investigated in vitro. Glioma cells go to apoptosis by glucose withdrawal, unlike normal astrocytes. Apoptosis is a cell death mechanism that allows the elimination of damaged cells. In cancer, including glioma, the dysregulation of apoptosis is related to aggressiveness. This process has been extensively reviewed elsewhere and we pointed out the relation with metabolism when is properly [46]. Mitochondrial respiration is over-activated to compensate for the decrease of ATP; however, ROS are also produced and lead to cell death in glioma cells, but not in normal astrocytes, even if the antioxidant capacity was the same in both [47]. We should consider that the metabolic effect of aerobic glycolysis could be more complex. Solid tumors are also heterogeneous and present gradients of oxygen and nutrients, and thus tumoral cells exhibit important metabolic adaptations. It has been documented that the glycolytic profile of glioma tumors could be different depending on the area of the tumor, being more glycolytic in the inner areas and with an OXPHOS profile in the outlier layers of the tumor [47,48]. 

Moreover, the enhanced glucose uptake of glioma cells seems to be related to metastasis and invasion, which is linked to poor prognosis. In this sense, the income of glucose through transporters as glucose transporter 1 (GLUT1) seems to be crucial. Miyai and coworkers found that the H3.3K27M histone acetylation signal induces the expression of GLUT 1 and was correlated with aerobic glycolysis and invasion in gliomas [49]. Moreover, the inhibition of GLUT1 expression through the silencing of TUBB4, an isoform of tubulin, reduces the proliferation and formation of glioma spheroids [50].

The surrounding tumoral microenvironment, including immune cells, is also modified by glycolytic alterations and the Warburg effect. An increase in glucose uptake, glycolysis, and lactate production combined with a decrease in oxidative phosphorylation has been observed in dendritic cells but also in T cells, macrophages, neutrophils, B cells, and natural killer (NK). Moreover, the lactate support and acidic environment decrease the immune response affecting monocyte differentiation and T-cell response [51], supporting the proliferation of glioma cells through the immune system (Figure 2).

### 2.2. Glycolytic Enzymes in Glioma

#### 2.2.1. Hexokinase 2

The overexpression of hexokinase 2 (HK2) has been fundamental to explaining the Warburg effect in gliomas. HK2 participates in aerobic glycolysis, and its expression has been linked to cell proliferation, angiogenesis, invasion, metastasis, apoptosis, and chemotherapy/radiotherapy resistance as well as worse prognosis in glioblastoma patients [52]. Also, it has been reported that the overexpression of HK2 in GBM patients correlated with an overall survival rate significantly lower compared to normal and low-grade glioma patients [53].

Normal brain expresses HK1, but in glioma, an overexpression of HK2 has been reported; in fact, HK2 usually is expressed in skeletal and adipose tissue, but not in the brain. When HK2 is downregulated in glioblastoma cells, the extracellular lactate diminishes, whereas the consumption of oxygen, uptake of glucose, and expression of GLUT1 increases. Then HK2 seems to participate in the high avidity of glucose from GBM cells allowing them to maintain a high rate of glycolytic activity even in the presence of oxygen contributing to the aggressiveness of GBM [52].

It has been proposed that HK2 is also regulated by microRNAs such as miR-542-3p. The overexpression of miR-542-3p increased the activity of HK2 and was correlated with poor prognosis [54]. In another example, the knockdown for mitochondrial PTEN-induced kinase 1 (PINK1) in glioma cells increased glucose uptake and lactate as well as hexokinase activity and the expression of glycolytic genes; this correlates with the low expression of PINK1 in glioblastoma and with low survival of these patients [55].

Moreover, it has been described that voltage-dependent anion channel (VDAC) contributes to the binding of HK2 to the outer membrane of the mitochondria; this disposition allows a better availability of the ATPs produced in the mitochondria and benefits glycolytic metabolism. For example, the positron emission tomography (PET) was established based on the VDAC/HK2 interaction. In PET, ^18^F-2-deoxyglucose (^18^F-2-DOG) is converted to ^18^F-2-DOG-6-P by VDAC bound to HK2, allowing the monitoring of cancer evolution in patients [56]. The VDAC/HK2 interaction has been tested in glioma models. For example, Gomisin J, a derivate of lignan, reduces the expression of HK2 and disrupts the bond between HK2 and VDAC, leading to apoptosis and tumor reduction in glioma [57]. Additionally, the downregulation of VDAC1 in glioblastoma models leads to an inhibition of tumor growth and reversion of malignity markers [58,59]. In another approach, VDAC1-based cell-penetrating peptides have been designed, which have domains and amino acid residues that target the interactions related to apoptosis, and then are capable of inducing apoptosis preferentially in cancer cells, as has been demonstrated in glioblastoma cells [60]. Furthermore, it has been suggested that HK2 suppresses the apoptosis in glioblastoma by blocking the mitochondrial translocation of the pro-apoptotic protein Bax and subsequently the release of cyt c from mitochondrial at cytosol [52,61]. Juang-Whang et al. suggested that mitochondrial HK2-bound VDAC inhibits the hydrolysis of Bid at tBid. The tBid induces the translocation from cytosol at mitochondria from pro-apoptotic Bax and Bak proteins, which negatively modulates at anti-apoptotic Bcl-2 and Bcl-_XL_ proteins promoting the release of cytochrome c into the cytosol and leads apoptotic cell death [62]. In this sense, HK2 depletion, but not of HK1 or PKM2 induces a reduction in the intracranial growth and angiogenesis of GBMs in vivo, with increased sensitivity to apoptotic cell death under hypoxia condition, radiation, and temozolomide through a downregulation in the HIF-1α stability and VEGF levels [52]. It has been suggested that the PI3K/AKT pathway, which is overexpressed in glioblastoma, induces growth-promoting effects of HK2 by both glucose phosphorylation and its mitochondrial translocation [63]. On the other hand, HK2 directly binds and inhibits mTORC1 leading to autophagy under glucose deprivation; moreover, this binding is suppressed by G6P, a product of HK2 enzymatic activity under glycolysis [64]. Autophagy is a catabolic process that can promote cell death or survival in cancer cells and the relevance of autophagy in glioma has been extensively revised previously [65].

On the other hand, it has been demonstrated that the RSL3 compound induces dysfunction in the glycolysis, which promotes autophagic cell death through inactivation in the HK2, PFK, and PKM2 enzymatic activities and increases autophagic markers such as pAMPK, pULK1, Beclin 1, Atg5,-12, and LC3II in human glioma cells as U251, U87, and U373, as well as xenograft glioma in vivo model. Moreover, the addition of sodium pyruvate inhibited the autophagy induced by RSL3 [66].

#### 2.2.2. Glucose-6-Phosphate Isomerase (G6PI)

Some glycolytic enzymes have been associated with bad prognosis; for example, glucose-6-phosphate isomerase (G6PI), which is induced by hypoxia in glioma cells. G6PI, also known as autocrine motility factor (AMF), performs enzymatic functions in the cytosol as a glycolytic enzyme [67] or if is secreted depending on the cellular context, as a cytokine to induce invasiveness by binding extracellularly to its gp78 receptor (AMFR) [68,69] activating a signaling pathway such as PI3K/AKT and then NF-κB and Wnt/β-catenin [70]. Also, AMFR positively regulates cell motility through the Rho-associated protein kinase 2 (ROCK2)/cofilin pathway [71] and inhibits apoptosis by downregulating the Apaf-1 and caspase-9 genic expression [72]. It has been reported that the transcript of AMF is elongated in GBMs compared with anaplastic astrocytomas and that the overall survival of GBM patients with AMF overexpression is lower than in glioblastoma patients without AMF [73]. Li and colleagues demonstrated that the silencing of G6P1/AMF suppresses the cell migration and tumorsphere formation as well as phosphorylation of AKT and SOX2, a stemness marker in U87 glioma cells. These data suggest that G6P1/AMF signaling regulates the cell proliferation of cancer stem cells [74].

#### 2.2.3. Phosphofructokinase

PFK is an important regulator of the glycolytic pathway [75]. At this point, glucose has been phosphorylated, avoiding its escape from the cell, and converted to fructose-6-phosphate. Then PFK1 allows the conversion to fructose-1,-6 biphosphate in an irreversible reaction spending ATP, indicating the fate of glucose to the glycolytic pathway. PFK-1 is strictly regulated; its activity is inhibited in the presence of ATP, the presence of citrate (from TCA), fatty acids, and acidic pH due the lactate production [76], allowing negative feedback in case of overactivation of the glycolytic pathway. On the contrary, another important regulator of PFK1 is fructose-2–6 biphosphate (F2,6BP) which activates PFK1, even in the presence of ATP [77,78]. PFK-2 converts fructose-1-phosphate to F2,6BP, which is degraded by F2,6BPase [76]. Interestingly, the bifunctional enzyme 6-phosphofructo-2-kinase/fructose-2,6-bisphosphatase (PFK-2/FBPase-2 or PFKFB) performs both, the formation and degradation of F2,6BP. Phosphorylation at Ser^32^ activates the biphosphatase and inhibits the kinase whereas the dephosphorylation of the same residue has the contrary effects [79]. The deregulation of this part of the glycolysis pathway has been extensively documented in cancer, either as PFK1 or PFKFB overactivation [80,81]. 

The first evidence of deregulation of PFK in gliomas reported less sensitivity to inhibition by citrate and, on the contrary, more susceptibility to the activation by F2,6BP [82]. Furthermore, PFKFB3 and PFKFB4 isoforms were considered markers of bad prognosis in patients with IDH-wild type glioblastoma [83,84]. Moreover, PFKFB4 has been associated with cell survival in brain cancer stem cells derived from glioblastoma patients: PFKFB4 silencing suppresses viability and inhibits the production of ATP and lactate [84]. Besides, transforming growth factor β1 (TGFβ1), upregulates the expression of GLUT1, HK2, LDHA, and PFKPB3. It has been reported that an increment in TGFβ1 secretion in glioblastoma affects cell metabolism [85]. Moreover, Smad, p38 MAPK, and PI3K/AKT signaling pathways seemed to be activated by TGFβ1 promoting the expression of PFKFB3 [86]. Interestingly, TGFβ1 is released by several immune cells of the tumoral microenvironment (TME) as tumor-associated macrophages (TAMs), particularly those with an M2 profile, and cancer-associated fibroblasts (CAFs) [87,88,89]. As we mentioned before, it has been documented that TGFβ1 induced the overexpression of PFKFB3 in glioma cells, leading to a glycolytic flux with an increase in the glucose uptake and lactate production [86].

Recently, other regulation points over PFK have been described. For instance, the voltage-dependent anion channel 2 (VDAC2) was downregulated in glioma stem cells (GSC), which have a high glycolytic profile compared to non-stem cell tumor cells (NSTCs). In addition, VDAC2 binds to the platelet type of phosphofructokinase (PFKP) and avoids its release from the mitochondrial membrane to the cytoplasm. The downregulation of VDAC2 and consequently the release and activation of PFKP lead to a glycolytic profile and benefit the transition to stem-cell phenotype. In patients, VDAC2 expression was inversely correlated with glioma grades; moreover, patients with VDAC2 expression had better survival than those with low VDAC2 expression [90]. 

#### 2.2.4. Aldolase

The function of the glycolytic enzymes is becoming more complex and affecting more than only the metabolism. For example, Aldolase A (Aldo A) is an important enzyme in glycolysis and other non-enzymatic functions have been documented in cancer [91]. Recently, the downregulation of ARST, a lncRNA, was reported in gliomas, and this was correlated with proliferation, migration, and invasion. More interestingly, ARST binds directly to Aldo A) and disrupts the interaction between Aldo A and actin. In that sense, exposed sites of actin are bound to cofilin which depolymerizes the cytoskeleton of actin and presents an alternative explanation of the tendency of gliomas to metastasis [92]. Also, it has been demonstrated that Aldo C co-localized with HIF-1, both are highly overexpressed in glioma cells whereas the adjacent tumor tissue showed a weak stain [93]. In addition, Stanke and colleagues reported a significant increase in Aldo C mRNA in patients with glioblastoma compared to normal patients [53]. On the other hand, Yu-Chang et al. proposed that Aldo C is a tumor suppressor associated with a significant increase in the survival of GBM patients [94]. Aldo C overexpression decreases the invasive capacity of U87 GBM cells through TWIST1 transcriptional factor downregulation, which regulates the epithelial-mesenchymal transition through positive genic regulation of IL8, 1L1, 1L1β, matrix metalloproteinase 1 (MMP1), and fibronectin [94].

#### 2.2.5. Glyceraldehyde 3 Phosphate Dehydrogenase

GA3PDH participates importantly in the Warburg effect catalyzing the reaction of GA3P to 1,3BPG in the glycolytic pathway [95]. It has been reported overexpression of GA3PDH in GBM human biopsies compared to gliomas of low-grade and normal tissue [96]. On the other hand, tumoral-associated macrophages (TAMs) phosphorylate and activate glyceraldehyde-3-phosphate dehydrogenase (GA3PDH) through the signaling with interleukin 1β (IL1β), inducing a glycolytic and proliferative state in glioma cells [97]. Glioma C6 and RG2 cells were treated with 4-phenylbutyrate, a histone deacetylase inhibitor, leading to the suppression of the GA3PDH transcription and therefore inducing a decrease in the cell proliferation and activation of the apoptosis by the inhibition of the glycolysis [96]. Nuclear GA3PDH is acetylated at Lys^160^ by the p300/CREB binding protein (CBP) acetyltransferases through direct bind, which in turn promotes the p300/CBP catalytic activity and the induction of apoptotic genes (*puma, Bax*) via p53 [98]. Moreover, the incubation of C6 glioma cells to moderate and severe hypoxia promoted the transcription of the *ga3pdh* gene, with the subsequent oxidation (inactivation) and accumulation of GA3DPH protein leading to the formation of GA3PDH/Hsp70 chaperone aggregates. The GA3PDH/Hsp70 aggregates inactivated the Hsp70 chaperone by its excessive occupancy, leading to the cell death of C6 cells [99]. Interestingly, the reduction of Hsp70 expression by triptolide inhibited its activity of chaperone on GA3PDH aggregation and induced apoptosis in C6 cells [100]. Hsp70 inhibits the caspase’s independent and dependent apoptotic pathway by neutralizing at Bax, tumor necrosis factor-related apoptosis-inducing ligand (TRAIL), and apoptosis-inducing factor (AIF) [101]. It has been described that localization (cytoplasm, mitochondria, endoplasmic reticulum, and nucleus) [102,103], stresses type (hypoxia, RNS, DNA, and damage) [104], conformational structure (tetrameric and homo-oligomeric) [105,106] as well as post-translational modifications (phosphorylation and acetylation) [107,108,109] modulates the localization and function of GA3PDH [72].

#### 2.2.6. Phosphoglycerate Kinase (PGK)

Phosphoglycerate kinase (PGK) 1 catalyzes the transfer of high-energy phosphate from the 1-position of 1,3-biphosphoglycerate (1,3BPG) to ADP, which leads to the generation of 3-phosphoglycerate (3PG) and ATP [110].

The overexpression of PGK1 mRNA in glioblastoma patients correlates with poor prognosis and overall survival [111]. In this sense, Li and colleagues reported that the overactivation of EGFR, KRAS, or BRAF in hypoxia leads to the activation of ERK signaling, which phosphorylate at Ser^203^ PGK1 with the subsequent mitochondrial translocation of PGK1 induced by PIN1 through TOM (translocase of the outer membrane) complex [112]. Mitochondrial PGK1 activates the pyruvate dehydrogenase kinase 1 (PDHK1) by phosphorylation at Thr^338^, which inhibits, also by phosphorylation, at the pyruvate dehydrogenase (PDH) complex; therefore, decreases the pyruvate concentration in mitochondria, increases of lactate levels, reduces the generation of reactive oxygen species, and stimulates the Warburg effect as well as promotion of the brain tumorigenesis [112]. Furthermore, a correlation between the phosphorylation levels in Ser^203^ PGK1 and Thr^338^ PDHK1 with the phosphorylation levels in Ser^293^ PDH and poor prognosis in GBM patients was reported [112]. Furthermore, the inhibition in the PKG1 activity by P7C3, a neurogenic compound, caused the suppression of the glycolytic process by decreasing the glucose uptake and pyruvate, lactate, and ATP levels as well as the cell growth of glioma in vitro and in vivo [113]. P7C3 binds at PGK1 through asparagine and lysine residues which promotes their degradation and low levels of the PGK1 protein [113]. Chen and colleagues suggested that P7C3 induces the degradation of PGK1 through of autophagy-lysosome pathway due to an increase of phagophores, autophagosomes, autophago-lysosomes as well as higher Beclin1 and LC3I protein levels [113]. Also, it was observed an overexpression of PGK1 mRNA induced by HIF-1 exposed at 2% and 0.2% O_2_ in U87, U373, and A172 in monolayer cultures and U87 gliomasphere cultures [114]. Qian and coworkers demonstrated that hypoxic conditions or glutamine deprivation results in the acetylation at Lys^388^ PGK1 by acetyltransferase NAA10. This acetylation promotes the binding of PGK1 and Beclin1 with the subsequent phosphorylation at Ser^30^ Beclin1 by PGK1, which enhances the formation and activity of Atg14L/VPS34/Beclin1/VPS15 complex. Subsequently, this complex increases the phosphatidylinositol-3-phosphate concentration to promote autophagy initiation in glioma human U87 cells contributing to brain tumor growth and aggressiveness [115]. However, under nutrient-rich conditions, the inactivation of mTOR kinase by the phosphorylation at Ser^228^ NAA10 leads to the inhibition of autophagy [116]. Furthermore, higher levels of acetylation at Lys^388^ PGK1 significantly correlate with the phosphorylation at Ser^30^ Beclin1 with poor prognosis in GBM patients. However, when acetylation and phosphorylation levels are low, overall survival duration more than doubles [115]. It has been demonstrated that Long non-coding RNA GBCDRlnc1 activated autophagy mainly by binding with PGK1 and inhibiting its ubiquitination, so the overexpression of PGK1 increased expression of Atg5/Atg12 conjugate [117]. In addition, it has been reported that PGK1 stimulates the radioresistance in human glioma cells, upregulates cell proliferation, and increases migration and invasion [118]. Additionally, it has been associated with high expression levels of Cofilin 1 and PGK1 with radioresistance in GBM patients [111,119,120], suggesting that Cofilin 1 and PGK1 can be used to evaluate glioma radiosensitivity and prognosis [120].

#### 2.2.7. Phosphoglycerate Mutase (PGM)

As we already commented, the PGM transfers a phosphate group from the C3 to C2 carbon to obtain 2-phosphoglycerate (2PG) as a product [121]. Interestingly, PGM1, a brain isoform, is highly expressed in C6 cells from rat glioma cells, high-grade human astrocytoma, and human glioblastoma samples [122,123]. Furthermore, the inhibition of PGM1 expression by siRNA led to a decrease in cell proliferation, invasion, and metastasis, while apoptosis and cell cycle arrest were induced [124]. 

Besides the effects of PGM1 on glucose metabolism, PGM1 regulates the ataxia-telangiectasia (ATM) kinase signaling, which allows the repair of DNA after DNA double-strand breaks (DSBs) [125]. Moreover, the overexpression of PGM1 in glioma has been correlated with an increase in the DNA repair damage mechanism related to ATM. In this sense, PGM1 sequesters the wild-type p53-induced phosphatase 1 (WIP1) in the cytoplasm, which usually inactivates the ATM pathway [123]. When the PGM1 expression is inhibited, glioma cells become less resistant to radio and chemotherapy [123,126]. Similarly, the knockdown of PGM1 resulted in a decrease in the cell viability of glioma cells; furthermore, in glioblastoma xenografts the knockdown of PGM1 led to an increment of 14.5% in the survival rate of the mice [127].

#### 2.2.8. Enolase

Enolase or phosphopyruvate hydratase transforms the 2-phosphoglycerate to phosphoenolpyruvate, the penultimate reaction of glycolysis producing an ATP molecule. A preliminary report showed that the serum levels of a tissue-specific isoform known as neuron-specific enolase (NSE), also known as ENO2, have not been correlated to a worse prognostic value in glioma patients yet [128]. However, in rat models, overexpression of ENO2 in glioma cells compared to normal cells was also reported by immunoradiometric and immunohistochemical assays [129]. ENO2 resulted to be a target of Cathepsin X, a cysteine peptidase that is overexpressed in glioblastoma. Moreover, Cathepsin X decreases the proliferative activity of α-enolase and the inhibition of Cathepsin X led to a decrease in the viability of the glioblastoma cells, being a potential therapeutic target for glioblastoma [130]. Another group reported overexpression of mRNA and protein levels of ENO1 in glioma samples. ENO1 induces the activation of the PI3K/AKT pathway, supporting the proliferation, migration, and epithelial-mesenchymal transition through the activity Cyc D1, Cyc E, and NF-κB, resulting in the hyperphosphorylation of Rb (inactivation) and upregulation of SNAIL and SLUG transcription factors, vimentin, and N-cadherin [131]. The overexpression of WW domain-binding protein 2 (WBP2) also has been detected in glioma; moreover, WBP2 was associated with the binding to α-enolase protein (codified by ENO1), which was associated with cell proliferation and migration, along with an increase in the glycolytic activity through ENO1/PI3K/AKT pathway [132]. In fact, α-enolase, the protein product of ENO1, is a multifunctional protein that has been associated with metastasis in several types of cancer, including glioma. It has been proposed that α-enolase is located in the cell membrane and acts as a receptor for plasminogen, which is converted to plasmin. Then, plasmin supports the degradation of the extracellular matrix, which is linked to metastasis [133].

Moreover, an overexpression of the long noncoding RNA (lncRNA) small nucleolar RNA host gene 18 (SNHG18) in glioma resulted in the suppression of the translocation of ENO1 to the nucleus, promoting the motility of glioma cells and thus metastasis due in part at cytoplasmatic ENO1; in fact, the transgenic suppression of ENO1 inhibits the invasive capacity induced by the upregulation of SNHG18 in glioma cells [134]. Therefore the authors suggest that the function of ENO1 in the glioma cells depends on ENO1 subcellular localization [134]. Also, the glycolytic activity of ENO1 has been correlated with the expression of ATP citrate lyase genic expression (ACL) in glioma cells, considering ENO1 as a metabolic tumor promoter [135]. Then, a rational search for ENO inhibitors has been made, directed to avoid glycolysis and tumoral growth in glioma. In this sense, it has been proposed an interesting mechanism known as collateral lethality. Taking advantage of deletions of ENO1 which can occur in some types of tumors including glioma, pan-ENO inhibitors were evaluated as SF2312 and also combined with pivaloyloxymethyl (POM) to form POMSF and POMHEX, a less toxic version of the inhibitor [136]. The treatment with POHMEX was used then as a successful and effective antineoplastic approach in ENO1-homozygously deleted glioma [136,137]. Another inhibitor known as ENO block seems to be effective as well; even its specific mechanism of action is still in discussion [138]. 

#### 2.2.9. Pyruvate Kinase

Pyruvate kinase converts PEP to pyruvate obtaining ATP in the process. It has been described as four isoenzymes. The L-PK (liver, kidney, and intestine) and R-PK (erythrocytes) are tissue-specific. The pyruvate kinase isoenzyme type M1 (PKM1) is expressed in practically all tissues; on the contrary, the PKM2 is expressed in high proliferating tissues such as lung, retina, embryonic cells, adult stem cells, and particularly in tumoral cells [139,140].

In glioma, it has been described a correlation between the expression of PKM2 and the grade of aggressiveness of the disease [140]. Furthermore, PKM2 has been related to a decrease in apoptosis through a mechanism that implies its translocation to the mitochondria, then phosphorylates at Thr^69^ Bcl-2, avoiding Bcl-2 degradation by Cul3-RBX1 ligase and in consequence, avoiding apoptosis [141]. In contrast, the ATPase activity of Hsp90α1 mediates the binding among Bcl-2 and PKM2. It has been observed that the tetrameric form of PKM2 activates the production of pyruvate; however, in a dimeric state, PKM2 is translocated to the nucleus acting as a transcriptional coactivator to induce the expression of pro-tumorigenic proteins via HIF-1β, STAT3, and c-myc [142]. The fructose-1,6-bisphosphate (F1,6BP) recognizes the dimers of PK2M and changes its conformation allowing the formation of the tetramer of PK2M. PKM2 also acts as a protein kinase; after the activation of EGF/EGFR, PMK2 is translocated to the nucleus and binds and phosphorylates at Thr^11^ histone H3 inducing the release of Histone Deacetylase (HDAC) 3 from histone H3 with the subsequent acetylation at Lys^9^ histone H3 and genic expression of *c-myc* and *cyclin D1*, which promotes the cell proliferation and tumorigenesis of glioma cells [143]. The levels of phospho Thr^11^ histone H3 and nuclear PKM2 are associated with the grades of glioma malignancy [143].

Additionally, nuclear PKM2, after activation of EGFR, is associated with transactivation with β-catenin at pTyr^333^, inducing the expression of cyclin D1 [144]. In this regard, it has been proposed that the PKM2/β-catenin complex promotes the transcription of c-myc and subsequently the GLUT1 and LDHA expression to induce the Warburg effect. Trametinib (MAPK inhibitor) binds and inactivates PKM2, reducing the levels of nuclear PKM through the suppression of the PKM2/c-myc axis, blocking the glycolysis, migration, and invasion in glioma cells [145]. In this sense, also it has been suggested that the phosphorylation in Tyr^105^ PKM2, induced by the chemokine receptor US28, promotes the translocation of nuclear PKM2 which acts as a transcriptional coactivator of HIF-1 and STAT3 transcription factors enhancing the metabolic reprogramming, cell proliferation, and angiogenesis in glioma cells [146]. Furthermore, it has been described that nuclear dimeric PKM2 induces the phosphorylation, dimerization, nuclear translocation, and transcriptional activity of STAT3 in cancer cells [142,147].

#### 2.2.10. Lactate Dehydrogenase

The LDH converts pyruvate to lactate and nicotinamide adenine dinucleotide (NAD)H) to NAD+, usually in anaerobic conditions, whereas NAD+ is required to convert glyceraldehyde-3-phosphate (GA3P) to 1,3 biphosphoglycerate to continue with the glycolysis. This option is less efficient than the NAD+ obtained from the electron transport chain; however, it is the fastest. Importantly the overexpression of LDHA and overproduction of lactate in gliomas and other tumors has been established as a potential biomarker of malignity [148,149]. Also, it has been observed that the predominance of glycolysis and lactate production has been used as a clinical marker of stem cell infiltration in gliomas, detected by magnetic resonance spectroscopy. High levels of lactate and LDHA were detected in highly invasive glioblastomas compared to low invasive GBM, which limited the tumor-infiltrating area [150]. It has been demonstrated that lactate induces migration and cellular invasion in high-grade glioma which overexpresses higher LDHA levels through transforming growth factor β2 (TGFβ2)/integrin α_v_β_3_/MMP2 signaling [151]. TGFβ2 is highly expressed in glioblastoma and has been correlated with a bad prognosis.

Interestingly, TGFβ2 is also associated with an increment in glycolysis in glioma and it is released also by several immune cells of the TME as TAMs, particularly those with an M2 profile, and cancer-associated fibroblasts (CAFs) [87,88,89]. On the other hand, the lactate flux can be mediated by monocarboxylate transporters (MCT) 1 and 4, which are overexpressed in glioblastoma linked to the worst prognosis [152,153,154,155]. When the expression of MCT4 was inhibited in stem-like glioblastoma neurospheres, an increment in apoptosis and cell death was observed [154,156]. Likewise, in an in vitro model of glioma stem cells using spheroids of U251, a p53 mutant cells, MCT1 levels increased, while its inhibition suppressed their proliferation [157]. The MCT1 and MCT4 both have an affinity to lactate and ketone bodies; however, MCT1 import while MCT4 export these substrates, modulating the lactate efflux in the cell and this tendency has been reported also in glioma [152]. 

It has been described that the LDH-A phosphorylated at Tyr^10^ and Tyr^83^ promote the Warburg effect and tumor growth by the modulation of the NADH/NAD redox homeostasis [158]. pTyr^10^ of LDHA promotes the tetrameric (active) LDHA formation, whereas Tyr^83^ promotes its binding to NADH [148,158]. It was reported that cyclin G2 has anti-tumoral properties in glioma, including the inhibition of glycolysis through the suppression of tyr^10^ LDHA inducing the reduction of the acidic tumoral microenvironment with a decrease in the infiltration of Tregs and antitumoral cytokines. Additionally, the same cyclin G2 potentiates the antitumoral functions of PD-1, being a potential molecule with action in glycolytic metabolism and immune response to cancer (Figure 2) [159].

**Figure 2 ijms-24-17633-f002:**
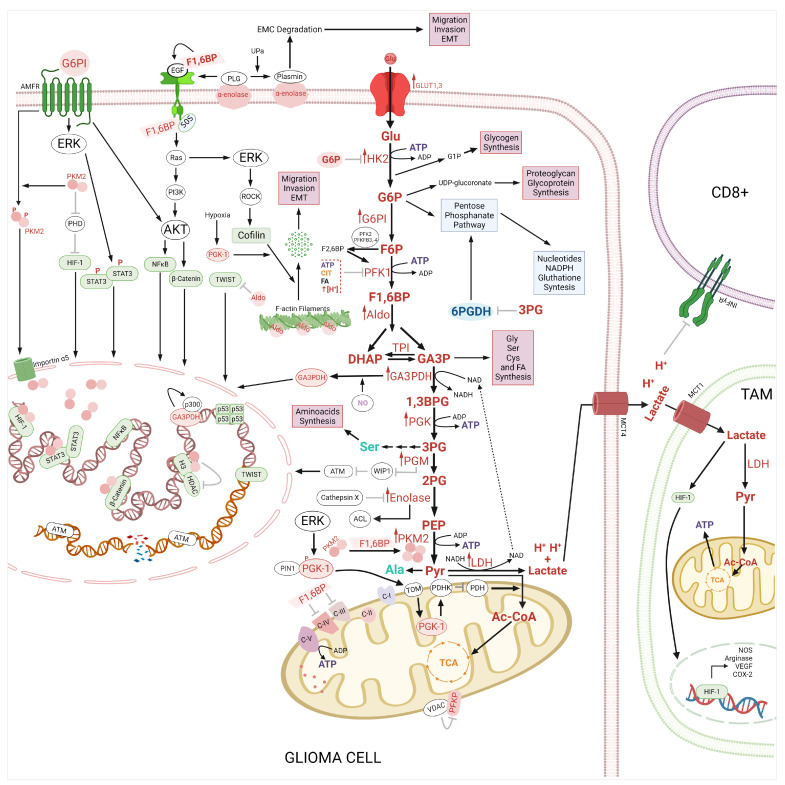
Role of glycolysis in glioma. Hexokinase 2 (HK) phosphorylates at glucose to obtain glucose-6 phosphate (G6P). G6P could be redirected to the pentose phosphate pathway (PPP) and at the synthesis of glycogen and UDP-glucuronate. G6PI binds to its autocrine motility factor receptor (AMFR) activating AKT and ERK kinases, which activate STAT3, NF-κB, β-Catenin and Cofilin. The VDAC2 binds to the platelet type of phosphofructokinase (PFKP) in the mitochondria inhibiting its activity. On the other hand, F1,6BP binds to EGFR and SOS blocking the EGFR/RAS/PI3K/AKT signaling pathway and inducing the formation of the tetramer of PKM2. In addition, F1,6BP inhibits the Complex-III and IV (C-III, -IV) from the electron transport chain (ETC). TPI induces the stabilization of the actin filaments and inhibits Twist-related protein (TWIST). PGK1 is phosphorylated by ERK and translocated to mitochondria via PTEN-induced kinase 1 (PIN1)/translocase of the outer membrane (TOM) complex, whereas Mitochondrial PGK1 activates the pyruvate dehydrogenase kinase 1 (PDHK1). PGM1 inactivates p53-induced phosphatase 1 (WIP1), which inhibits ATM. 2PG is metabolized to phosphoenolpyruvate (PEP) by enolase (ENO), which is inhibited by Cathepsin X. ENO induces the activation of the PI3K/AKT pathway and the expression of ATP citrate lyase (ACL). ENO is also a receptor for plasminogen (PLG), which is converted to plasmin. The nuclear PKM2 acts both as a protein kinase for histone H3 and as a transcriptional coactivator for hypoxia-inducible factor-1 α (HIF-1α), STAT3, and β-Catenin. The lactate and H+ are expelled by the monocarboxylate transporter 4 (MCT4), acidifying the tumor microenvironment, and promoting the death of cytotoxic (CD8+) T cells. Furthermore, the lactate is a metabolic fuel for tumor-associated macrophages (TAM) and induces nitric oxidase synthase (NOS), arginase, vascular endothelial grown factor (VEGF), cyclooxygenase (COX-2), and overexpression via HIF-1. Abbreviation: adenosine triphosphate (ATP), citrate (CIT), phosphofructo-2-kinase/fructose-2,6-biphosphatase 3,-4 (PFKFB-3,-4), pyruvate dehydrogenase (PDH), HIF-1α prolyl-hydroxylases (PHD) and Rho-associated protein kinase 2 (ROCK2). ↑ means overexpression, ↑ activation, ⊥ and inhibition. The figure was created with BioRender.com.

### 2.3. Therapeutic Implications of Glycolysis and Lipid Metabolism in the Apoptosis Regulation in Glioma

Glycolysis is related to other metabolic pathways, including lipid metabolism as we briefly describe in Figure 1. Besides, It has been proposed that resistance to cell death in glioma cells is due to the dysregulation of apoptotic and autophagic pathways through the overexpression for AMPK, TKR/PI3K/AKT/mTOR, p53, and HIF-1 signaling pathway [160]. AKT can modulate both processes by inducing glycolysis and lipids metabolism, which supply the energetic and cellular components for proliferation and cellular survival under several stress conditions [161]. 

On the other hand, it has been described that the glycolytic pathway modulates apoptosis via PMK2, GA3PDH, and HK2 [162] (Figure 2 and Figure 3), whereas HK2 and PMK2 inhibit the apoptosis in cancer cells to confer cellular protection under starvation conditions such as glucose and glutamine promoting the migration and high resistance to therapies [163,164]. Dong-Qiang et al. demonstrated that 3-Bromopyruvate (3BPr) induces apoptosis in CD133+ U87 glioma cells by increasing the Bax levels and the caspase3 activity [165]. 3-BPr recruits pro-apoptotic BAX protein to mitochondria promoting the inhibition of Bcl-2, and Mcl-1 and the caspase-3 activity [39,166,167]. 3-BPr blocks the binds of HK2 to the mitochondrial membrane and in turn, promotes the release of mitochondrial AIF to cytosol inducing cell death [167]. It has been suggested that the combination of 3-BPA with chemotherapeutic drugs can increase the chemoresistance in cancer cells by depleting the ATP, lipids, protein, nucleotides levels, and the DNA repair systems [168]. The administration of 3-BPr has been also evaluated in stage IV metastatic melanoma and patients with advanced hepatocellular carcinoma [169,170]. Lonidamine, another inhibitor of HK2 was administered in clinical trials for the treatment of several types of cancer including ovarian, lung, and breast cancer [171,172,173,174]. However, the hepatic and pancreatic toxicities presented by lonidamine have limited its clinical success [175].

Also, the lipids play an important role in the regulation of apoptosis in both the intrinsic and extrinsic pathways (Figure 3). In the intrinsic pathway, the regulation of apoptosis is carried out by mitochondrial lipids such as cardiolipin (CL), ceramide, and sphingosine-1-phosphate [176,177]. Cardiolipin induces the formation of Bax and Bak1 oligomers, for the formation of pores and consequently the release of cytochrome C into the cytoplasm, which induces the activation of caspases [178,179]. Cardiolipin has been reported to interact with Bidt by translocating to the mitochondrial outer membrane, thereby inducing the release of cytochrome C into the cytosol. Sorice et al. reported that glioma cells stimulated with Fas-induced the activation of caspase-3, the formation of CL/Bid complex [180]. Mitochondrial ceramide (Ce) is a lipid that is produced upon pro-apoptotic stimuli and represents the main requirement for Bax oligomerization, thus forming ceramide channels [181]. Glioma cells treated with indomethacin presented apoptosis with an increase in the ceramide generation and protein phosphatase A2 (PPA2), AKT inactivation, translocation of Bax from the cytosol to mitochondria, caspase 3 activation and down-regulation of Mcl-1. The apoptotic events were also duplicated with the administration of C2-ceramide andLY294002 (Akt inhibitor) [182]. It has been demonstrated that TMZ and radiation induce the sphingomyelase activity, which hydrolyzes sphingomyelins to ceramides inducing apoptosis [183]. However, cells can transform ceramides sphingosine-1-phosphate to evade apoptosis [183]. 

On the other hand, sphingosine-1-phosphate (S1P) upon degradation by sphingosine-1-phosphate lyase 1 forms hexadecenal, which interacts with Bak1 and Bax respectively, inducing oligomerization and cytochrome C release [184]. Amaegberi et al. demonstrated that 2-hexadecenal (HexaD) inhibits cell proliferation and induces apoptosis in glioma cells [185]. Cholesterol also plays a role in apoptosis since the presence of cholesterol inhibits Bax and contributes to the decreased cellular ability to induce mitochondrial outer membrane permeabilization. In addition, cholesterol can inhibit the insertion and activation of Bax, thus preventing apoptosis [186,187]. Studies have reported the possible role of polyunsaturated fatty acids (PUFA) in cancer cell apoptosis through Bcl-2, cytochrome C, and caspases [188,189]. Oleic acid is an anticancer agent that induces autophagy and apoptosis. The role of stearic acid as an inhibitor of invasion, proliferation, and inducer of apoptosis in cancer cells has also been studied, however, the mechanisms remain unclear [190]. Conjugates of temozolomide with fatty acids such as palmitic acid, linoleic acid, and oleic acid increase the enhancer TMZ efficacy, overcoming the resistance at TMZ induced by MGMT glioma models in vitro and in vivo [191]. Furthermore, it has been demonstrated that enzymes involved in the lipid metabolism regulate the apoptosis. FASN inhibits apoptosis by suppressing the Puma, Bax, and Noxa pro-apoptotic proteins [192]. FASN inhibitors induce apoptosis in glioma cells by activation of caspases 3 and downregulation of Bcl-2 protein [193]. These studies suggest that drug sensitivity in glioma cells could be enhanced through inhibitors from glycolysis and lipid metabolism in combination with an activator of the apoptosis pathway.

### 2.4. Regulation of the Autophagy by Glycolysis and Lipid Metabolism

Autophagy is a catabolic process that can promote cell death or survival in cancer cells and the relevance of autophagy in glioma has been extensively revised previously [65]. It has been reported that the enzymes and metabolites related to glycolysis participate in the regulation of autophagy [161,162] (Figure 4).

Autophagy is a highly conserved and regulated process to digest organelles and proteins and recycle cytoplasmatic components, which can protect the cancer cells from harsh environments increasing their capacity for adaptation and survival [194]. In cancer cells, autophagy can increase or decrease with chemotherapy agents playing an essential role in cell death (suppressor tumoral) or cell proliferation (promotor tumoral) depending on the cellular context. Therefore, the modulation of the autophagy is an attractive target for anti-tumor therapies.

Glioma cells present high rates of drug resistance, and autophagy contributes to the resistance to treatment. TMZ is generally used as therapy for the first line of GBM, however, TMZ resistance is modulated by autophagy [195]. Also, it has been observed that bevacizumab (VEGFR inhibitor) also shows low efficacy on glioma cells due to the promotion of autophagy in hypoxic conditions [196]. Under hypoxia, HIF-1 increases the Bcl-2 interacting protein 3 (BNIP3), which is released at Beclin 1 from Becli1/Bcl-2 complex inducing autophagy in U87 and T96G) cells. Bevacizumab plus chloroquine increases the glioma cell response to treatment [196].

Autophagy is generally induced due to a diminution in the ATP levels, nutrient deprivation, and hypoxia inducing the AMPK activation and in turn the inhibition of mTOR [161]. Glucose deprivation induces autophagy through HK2, which binds at mTOR inhibiting its activity [64]. mTOR inhibits autophagy by negatively regulating the activation of unc-51-like autophagy activating kinase 1 (ULK1) complex, which is necessary for the phagophore formation [64]. Then, it has been suggested that a HK2 molecular link between glycolysis and autophagy ensures cellular energy supply [64]. The impact of metformin as a major regulator of glucose metabolic pathways has been extensively reviewed elsewhere [197]. Moreover, Sesen et al. demonstrated that metformin (HK2 inhibitor) promotes apoptosis and autophagy by increases at LC3B and Beclin 1 and decreases sequestosome 1 (SQSTM1) levels in glioma cells by activating Redd1 (TSC2 activator) and AMPK and inhibiting mTOR, S6K and 4EBP1 increasing the cytotoxic effects of radiotherapy and temozolomide [198,199]. A combination of arsenic trioxide plus metformin promoted autophagy and apoptosis in glioma cells through the activation of AMPK/FOXO3 [200]. FOXO3 regulates the transcription of autophagic genes such as ULK1/2, Beclin1, Atg4,-5,-8 and-14 [65,160,197].

Nuclear PFKFB3 induces autophagy by phosphorylating at AMPK. On the other hand, the suppression of PFKFB3 by 3-(3-pyridinyl)-1-(4-pyridinyl)- 2-propen-1-one in HCT-116 cells induces autophagy as a pro-survival mechanism through the ROS generation, decrease in glucose levels and inactivation of p70S6K and S6 [201]. Blum et al. demonstrated that pharmacological inhibition of Ras in glioma cells caused a reduction in HIF-1α expression via inhibition of the PI3K/AKT pathway and in turn decreases of PFKFB3 and glycolysis, promoting cell death [202]. On the other hand, it has been reported that under glucose starvation AMPK phosphorylated at GA3PDH inducing its nuclear translocation, whereas to bind at Sirt1 histone deacetylases inducing a rapid initiation of autophagy [203]. Sirt1 can bind and activate to Atg5, atg7, and atg8 autophagic complex [204]. Also, it has been reported that GA3PDH inhibits mTOR by binding at Rheb [205]. Dando et al. reported that the inhibition of mitochondrial uncoupling protein 2 (UCP2) protein induces the ROS generation and in turn nuclear translocation GA3PDH, inactivation of Akt/mTOR/P70S6K signaling pathway promoting the apoptosis independent of caspases and autophagy mediated by Beclin1 in pancreatic adenocarcinoma cells [206]. PGK1, another glycolytic enzyme, also modulates autophagy. Hypoxia or glutamine starvation in glioma cells promotes the phosphorylation of Ser301 Beclin1 by PGK1 through the mTOR inhibition which promotes the activation of autophagic initiation complex, which contributes to the brain tumoral aggressiveness [115]. PGK1 promotes radioresistance and invasion in glioma cells.

In addition, several studies demonstrated that PKM2 modulates autophagy. It has been reported that PMK2 activates autophagy by inhibiting phosphorylation at AKT1 substrate 1 (AKT1S1) a mTORC1 inhibitor, which facilitates mTOR activation [207]. It has been reported a positive loop between PKM2 and mTORC1, that mTORC1 increases PKM2 expression and in consequence induces an accumulation and utilization of metabolic intermediates [207]. PKM2 knockdown lung cancer cell lines increase the radiosensitivity at ionizing radiation by promoting apoptosis and autophagy [208]. PKM2 also induces survival mechanism autophagy by the phosphorylation at Thr119 Beclin-1 [209]. pThr119 Beclin-1 is released from Bcl2/Beclin1 complex for induces autophagy. On the other hand, it has been proposed that Beclin1 could show a pro-apoptotic effect by preventing of function of Bcl-2 and BclxL in glioma cells [210]. The PMK2 overexpression by histone methyltransferase G9a inhibition in glioma cells modulates the LC-3II and YAP-1 levels promoting autophagy [162]. Another study demonstrated that LDHB promotes autophagy by contributing to lysosomal acidification and formation of the autophagosome and thus cancer cell survival [211].

Also, autophagy could be modulated by several lipid species such as sterols, sphingolipids, and phospholipids derived from glycolysis intermediaries (Figure 4). These lipids and lipid-related processes act on signaling processes throughout all stages involved in autophagy such as initiation, autophagosome development, and autolysosome maturation. Phosphoinositides act as key modulators of autophagy [192]. PtdIns(3,4,5)P3 participated in the regulation of autophagy by activation of the mTOR signaling pathway [192,212]. The phagophore is formed from PtdIns3P-enriched sites, that allow the recruitment of diverse PtdIns3P- binding proteins [212]. PtdIns5P synthesis is required for autophagosome formation [213]. PtdIns4P reduces the autophagic flux by blocking lysosome and autophagosome fusion [214].

Sphingolipids are other molecules that participate in the induction of autophagy [214], one of them is the ceramide. Due to increased ceramide levels, JNK/c-JUN promotes autophagy through MAP1LC3 and Beclin-1 transcription [215,216]. Also, it has been demonstrated that fatty acids regulate the autophagy. In this sense, omega-3-polyunsaturated fatty acids (ω3-PUFAs), such as docosahexaenoic acid (DHA), have been reported to increase autophagy-like cell death mechanism by activating AMPK and dephosphorylating AKT and mTOR in glioma cells [217].

On the other hand, fatty acid synthase (FASN) FASN inhibits the autophagy through mTOR [218]. mTOR inactivates by phosphorylate at TFEB promoting its cytoplasmatic localization [218]. FASN is overexpressed in human glioma samples and its inhibition by Orlistat induces autophagy in glioma [219]. FASN inhibitors, also significantly reduce the tumor volume and angiogenesis by downregulation of VEGF and HIF-1α [220]. Simvastatin a hydroxymethylglutaryl-coenzyme A reductase (HMG-CoAR) inhibitor promotes cell death in C6 and U251 glioma by inducing the AMPK, LC3-II Beclin1 activation, inactivation of AKT and in turn mTOR/p70 S6 kinase1, also down the mevalonate levels [221]. On the other hand, Sphingosine-1-Kinase (SPHK1) modulates the autophagy and then regulates the growth, invasion, and therapy resistance, cancer development, growth, and metastasis [222]. SPHK1 induces autophagy through of inhibition of mTOR, and up-regulation of ULK1 and Atg proteins in cancer cells [161].

Then, to increase the drug sensitivity on glioma cells could be necessary therapy that involves inhibitors of the cell metabolism in combination with activators or suppressors of the autophagy pathway.

### 2.5. Glycolysis as a Target of Therapeutic Drugs in Gliomas

It is evident the importance of glycolysis in glioma, and subsequently, more alternatives that regulate glycolysis and the Warburg effect have been proposed to treat this disease. For instance, methylene blue increased the cellular oxygen consumption rate (OCR) and decreased extracellular acidification rate (ECAR) as well as lactate production in glioblastoma cell lines [223], suggesting a possible treatment with this compound usually used in diagnosis (Table 1). 

Another research group developed enolase inhibitors called HEX, and its pro-drug POMHEX. In a very interesting approach, the authors propose collateral lethality, meaning that the loss of metabolic genes could be useful because the redundant isoform of the gene is quite susceptible to responding to the inhibitors. Then, they used gliomas with deletion of ENO1, but sensible to enolase inhibitors because of the expression of ENO2. The enolase inhibitors decrease in the TCA cycle metabolites with an evident increment in vitro; however, their efficacy in vivo was restricted suggesting that barriers such as the blood-brain barrier could avoid the recruitment of the drug to the brain [136].

The Warburg effect seems to benefit radioresistance cells in GBM. Then, another approach proposed dichloroacetate (DCA) as a possible alternative treatment. DCA has been used before as a treatment for congenital mitochondrial diseases in children and inhibits the pyruvate dehydrogenase kinases with activation of oxidative phosphorylation [224]. However clinical trials still have no conclusive results.

Inhibitors directed to PFKFB3, such as 3PO, have also been proven in glioma models. For instance, treatment with 3PO and bevacizumab, a monoclonal antibody against VEGF, decreased cell proliferation and increased apoptosis in vitro, whereas delayed tumor growth and improved survival using in vivo assays [226].

Another approach includes the use of an anticancer compound library searching for new anticancer drugs. For instance, R406, a spleen tyrosine kinase (Syk) inhibitor for immune thrombocytopenia, was proven in glioma. Interestingly R406 shifts the glioma cells from a glycolytic profile toward OXPHOS metabolism, resulting in an increment of apoptosis [227].

MCT4 also could be used as a therapeutic target in glioblastoma. In this sense, Acriflavine is a potent inhibitor that disrupts the interaction of MCT4 with CD147 (or Basigin), a cell surface glycoprotein, inducing a reduction in cell proliferation and tumoral growth [228,229]. Moreover, to induce lactate accumulation, *MCT4*, and *BASIGIN* genes were disrupted, and then an MCT1 inhibitor was used, leading to lactate accumulation, decreasing glycolysis and redirecting the metabolism to oxidative phosphorylation. Furthermore, glioma cells with MCT4/CD147 knockout result more receptive to the treatment with phenformin, a mitochondrial complex I inhibitor, apparently because the tumor cells that survive MCT blockade enter into an “ATP crisis” and die, implying that the use of MCT inhibitors could be a potential therapeutic perspective [243]. Besides, MCT1 and MCT4 seem to be a target of Vorinostat, an HDAC inhibitor, resulting in a 30% reduction of intracellular lactate; affecting the metabolism of glioblastoma cells reflexed by a 40% decrease in the hyperpolarize hyperpolarized [1-13C] lactate measure by magnetic resonance spectroscopic imaging (MRSI), which in turn decreases the cell viability and tumor progression [230]. Interestingly, glioblastomas with IDH1 mutation have low levels of MCT1 and MCT4 and present no changes in hyperpolarized [1-13C] lactate in MRSI [244,245].

Nonetheless, more challenges appear on the horizon. GBM could have a heterogeneous cell population inside the same tumor. Some evidence suggests that the subpopulation of glioma stem-like cells (GSC) in the tumors could have a quiescent metabolic profile and thus, be resistant to the inhibitors of glycolysis [246]. On the other hand, as we noticed previously, other groups had reported a high dependence of the glycolytic pathway in GSC and adaptation to a hypoxic microenvironment [48,90,247].

Moreover, GBM has been classified into four molecular types: classical characterized by EGFR amplification and CDNM2A deletion, mesenchymal (mesenchymal markers with NF1 deletion), proneural (PDGFRA amplification and isocitrate dehydrogenase 1 (IDH) mutation), and neural (neuronal markers) [248]. Interestingly, the mesenchymal subtype has the major upregulation of the glycolytic pathway [224]. Another group used different GBM cell lines finding that those with a glycolytic profile as GBM38 respond to inhibitors such as DAC or metformin hydrochloride (MF), in contrast with cell lines with an oxidative profile that presented resistance (GBM27) [249].

## 3. Glutaminase Pathway

Glutamine (Gln) is the most abundant non-essential amino acid in the body [250]. In the cells, glutamine is an important precursor for the synthesis of nucleotides, amino acids, and proteins, and also provides NADPH (nicotinamide adenine dinucleotide phosphate) and GSH (glutathione) for redox homeostasis [251,252]. Glutamine also drives the uptake of the essential amino acids, activates the mammalian target of the rapamycin (mTOR) pathway, maintains the pH homeostasis via the NH_3_/NH_4_^+^ and the interorgan nitrogen exchange via ammonia (NH_3_) [253,254]. In mammals, glutamine is an important link between the carbon metabolism of carbohydrates and proteins [255,256].

Glutamine is produced by glutamine synthetase (GS) partly from the amidation of glutamate (Glu) by ammonia derived from purine metabolism and/or taken up from the circulation [257].

In the healthy brain, glutamine is the precursor of glutamate, an excitatory neurotransmitter and a precursor of the main inhibitory neurotransmitter γ-aminobutyric acid (GABA) [258]. Glutamine synthetase is exclusively expressed in astrocytes, making responsible astrocytes for glutamine synthesis and ammonia detoxification [259]. Glutamine deamination via GLS produces glutamate the prevalent excitatory neurotransmitter in the human brain. After glutamate is released from neurons is taken by the astrocytes via glutamate transporters, glutamate transporter 1 (GLT1) and glutamate/aspartate transporter (GLAST). In astrocytes, glutamate is amidated to form glutamine by GS. Then glutamine is transported outside de astrocyte by system N transporter, SN1 to finally be taken from the synaptic cleft by neurons, completing the glutamine-glutamate cycle [260,261]. Neurons are unable to synthesize either the neurotransmitter glutamate or the inhibitor neurotransmitter (GABA) from glucose, glutamate synthesis involves neuron-astrocyte cooperation termed the glutamine-glutamate cycle [262,263].

In cancer cells, excessive growth, proliferation, and survival require high amounts of energy, in adverse conditions of hypoxia, nutrient shortage, and immunological pressure of the host. This high energy demand is obtained from the glucose and the amino acid glutamine metabolism [264,265]. The reprogrammed metabolism of neoplasms adapts them to specific growth requirements and conditions that involve addiction to glucose (Warburg effect) and/or glutamine. Even in the presence of oxygen, due to its glucose addiction cancer cells consume high levels of glutamate and secrete high levels of lactate [266].

During the anabolism of glutamine, the glutamate formed is associated with different molecules involucrate such as αKG, lactate, and GSH [267]. GSH is a tripeptide comprised of three amino acids (cysteine (Cys), glutamate, and glycine (Gly)) and is an important antioxidant agent in the central nervous system (CNS), increased GSH levels are responsible for treatment resistance and essential for tumor development in gliomas [268]. In the glioma surrounding tissue, extracellular glutamate levels were found to be elevated compared to unaffected brains [269]. Glutamate release from glioma was confirmed in vivo through glioma cells implanted into rat brains [270]. High levels of extracellular glutamate can result in the production of reactive oxygen species (ROS) causing macromolecule damage, resulting in cell death [271]. Oxidative stress induces expression of the system xCT antiporter to synthesize GSH, via NRF2 activation (regulator of redox homeostasis) [272]. Chung and colleagues have demonstrated a prominent expression of xCT in biopsies from glioma patients and in biopsies obtained from nonmalignant brain tissue. Moreover, System xCT has been identified as the only cystine transporter expressed in different glioma cell lines [273]. The expression of NRF2 was significantly higher in glioma tissues than in normal brain tissues [274] and Interestingly PI3K/AKT axis is also shown to produce GSH by nuclear factor erythroid 2–related factor 2 (NRF2) stabilization [275,276]. Furthermore, the glutamine is converted to lactate and released from the cell, minoring the glycolytic Warburg effect. It has been demonstrated in the SF188 glioblastoma cell line that 60% of glutamine was finally converted to lactate [265].

### 3.1. Glutaminase Pathway in Glioblastoma

Malignant gliomas exhibit increased glutamine uptake and consumption (Figure 5) [277,278]. In this context, for instance, human GBM xenografts in mice showed an increase in the total pool of glutamine compared to the surrounding normal mice brain tissue [279]. This elevation in the intratumoral glutamine levels relative to the contralateral normal brain tissue is detected also in patients with GBM by magnetic resonance spectroscopy [280]. Gln transports at the glioma plasma membrane are induced compared to normal astrocytes to facilitate Gln entry and further catabolism (Figure 2) [278]. Glutamine transport via alanine/serine/cysteine transporter 2 (ASCT2) and Sodium-coupled neutral amino acid transporter 3 (SNAT3) is increased in glioblastoma [281,282]. In normal rat astrocytes in situ, glutamine transport is mediated via SNAT3 regulated by pH and sodium gradients. Moreover, the expression of sodium-dependent transporter ASCT2 is increased in C6 glioma cells [281,283]. Studies in human brains found that SNAT3 expression is increased in GBM tissues compared to lower-grade glioma and normal brain, suggesting that SNAT3 expression is directly related to the malignancy of the glioma [282]. The transport of glutamate has also been modified in gliomas. Studies in high-grade gliomas demonstrated that GLAST expression is enhanced compared to low-grade tumors, while GLT1 is decreased in glioma cells, enhancing extracellular glutamate levels [267,284]. Three transport systems are participating in the efflux of glutamine in cultured astrocytes, ASCT2, system L 1 (LAT1), and LAT2 [285] A portion of intracellular glutamine is released to extracellular space via LAT1, in exchange with leucine (Leu), this amino acid antiport activates mTOR synthesis and signaling [286]. In specific LAT1 is highly expressed in human gliomas and is directly related to glioma progression and poor prognosis in glioma patients [287,288]. The use of inhibitors to system L exerts cytotoxic effects by increasing apoptosis and decreasing proliferation in glioma cells with high LAT1 expression, [289]. Both, the ASCT2, LAT1 expression, GS, and GLS activity are upregulated by the activation of the oncogenic transcription factor c-myc [267,290,291]. Oncogene c-my is reported a poor prognosis in glial tumors [292]. Studies in SF188 glioma cells have proposed that c-myc regulates mitochondrial glutaminolysis and leads to glutamine addiction [267,293] (see Figure 2).

Although glutamine is a non-essential amino acid, most cancer cells cannot proliferate or survive in a medium without glutamine [294]. SF188 glioma cell line viability decreased to zero when glutamine was absent in the culture media, despite the presence of high levels of glucose [293,295]. On the other hand, the high energy needs of cancer cells to continuously proliferate, make them depend on glutamine as a source of nonessential amino acids [294]. The majority of intracellular aspartate came from glutaminolysis. Aspartate participation is required for proliferation partly because it supports nucleotide biosynthesis and, thus is a key determinant of cancer cell survival under glutamine-deprived conditions [295,296]. Other metabolites such as asparagine, alanine, and phosphoserine are synthesized from glutamine-derived nitrogens and are important for tumor cell survival [297,298]. In particular, asparagine has protective effects on glioblastoma tumor cells in glutamine-limited conditions by the activation transcription factor (ATF), preventing cell death. The asparagine synthetase is responsible for the asparagine synthesis from aspartate and glutamine and is a transcriptional target of ATF4. The activation of the asparagine synthetase (ASPS) correlates with the progression of the disease and poor prognosis of glioma and neuroblastoma patients [299]. 

Under normal conditions, 10–25% of acetyl-CoA for de novo lipogenesis comes from glutamine, although in cells under hypoxia, 80% of acetyl-CoA for lipogenesis comes from glutamine [300]. In glioblastoma cells, glutamine metabolism provides a carbon source facilitating the use of glucose-derived carbon and TCA cycle intermediates as biosynthetic precursors [301,302] and high levels of the GS enzyme have been detected by immunohistochemical analysis of GBM clinical samples.

#### 3.1.1. Glutamine Synthetase

GS high expression has been observed in gliomas [303]. However, GS expression varies between tumors according to a glutamine-rich tumor microenvironment, alleviating the need to synthesize glutamine, and suggesting a random loss of GS expression during neoplastic transformation or heterogeneity in their cellular origin [44,279]. A human tissue array of 150 glioma tissues showed that GS expression levels are higher in glioblastoma than in low-grade glioma or oligodendroglioma (OD) [279]. High levels of the GS enzyme have been detected by immunohistochemical analysis of GBM clinical samples [304]. GS expression is associated with poor prognosis in glioblastoma patients, correlated with high aggressiveness and worse survival [305]. Additionally, Rosati and colleagues found a correlation between low levels of GS in tumor samples and the presence of epilepsy [306].

GS activity fuels nucleotide biosynthesis and supports the growth of glutamine-restricted glioblastoma. During glutamine starvation, GS supplies glutamine for de novo purine biosynthesis in glioblastoma cells [304]. In glutamine, deprived C6 glioma cells have increased expression of GS and use ammonia as a substrate for glutamine de novo synthesis [307]. A marked decrease in circulating glutamine did not affect tumor growth in mice brains due to the circulation providing minimal amounts of glutamine to normal brain and glioblastoma [304,308], thus the glutamine necessary for the growth of GBM tumors induced in mice brain, is mainly synthesized by GS positive glioma cells or supplied by astrocytes. In the mitochondria, glutamine is converted by GS to glutamate, which is further catabolized to α-ketoglutarate to generate ATP through the production of NADH [309]. The silencing of GS decreased GBM cell line proliferation and colony formation both in the presence and absence of glutamine [304], whereas in another study the silencing GS potentiated rat C6 glioma cell motility [310] supplying NADH and amide group for fatty acid, sugars nucleotides synthesis, required in highly proliferative cells such as gliomas [267,301,311,312].

#### 3.1.2. Glutaminase

An abundance of glutamine accelerates glioma anabolism [267]. Glutamine may be converted to glutamate by GLS, which has two GLS1 and GLS2 isoforms [313,314]. GLS1 is associated with glutamine addiction in tumors and has oncogenic properties, while GLS2 is described as a context-dependent tumor suppressor factor [315,316]. It has been suggested that different tumors might have differential requirements of glutaminolysis. Upregulation of GLS1 enzymatic activity correlates with poor disease outcomes in patients with brain tumors [265]. The expression of the GLS1 gene is induced by c-myc associated with cell proliferation [317]. Pharmacological inhibition of GLS eradicates glioblastoma stem-like cells (GSCs), a small cell subpopulation in glioblastoma (GBM) responsible for therapy resistance and tumor recurrence [318]. GLS1 silencing or GLS2 overexpression plus an oxidative insult decreased proliferation in glioma cells [319]. The combination of PP242, a mTORC1 inhibitor, in combination with GLS inhibition (compound 968) in glioblastoma blocked tumor growth in tumor-bearing mice demonstrating that GLS inhibition reverses mTORC1- targeted therapy resistance [280].

**Figure 5 ijms-24-17633-f005:**
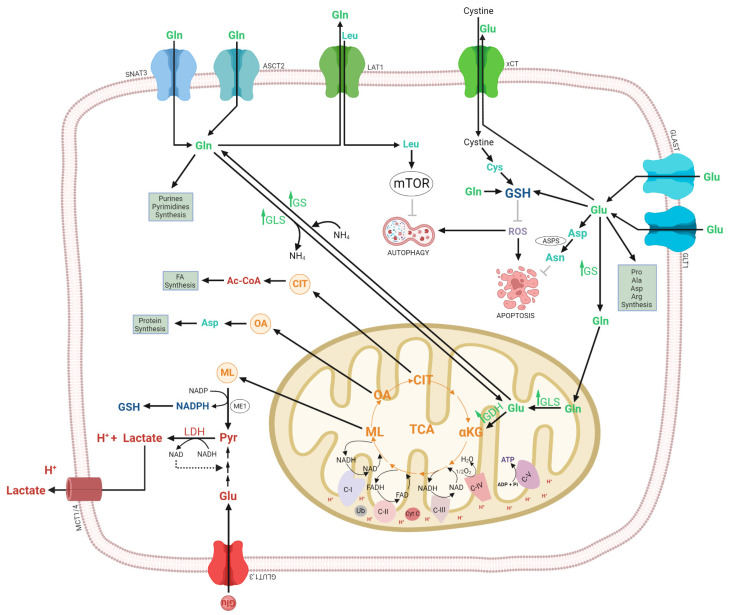
Glutamine pathway in glioma. The Glutamine (Gln) transport inside of cells by alanine/serine/cysteine transporter 2 (ASCT2) and system N transporter 3 (SNAT3) and its efflux is via LAT1 in exchange with leucine (Leu); this amino acid activates mammalian target of rapamycin (mTOR) synthesis and signaling. The Cytosolic Gln is transported to the mitochondrial and transformed to glutamate (Glu) by the glutaminase (GLS), and glutamate is converted into α-ketoglutarate (αKG) by glutamate dehydrogenase (GDH). αKG enters the TCA cycle and originates intermediates such as malate (ML), oxaloacetate (OA) and citrate (CIT), precursors of pyruvate (Pry), aspartate (Asp), acetyl CoA (Ac-CoA) and NADH and FADH, which induces the generation of glutathione (GSH), the acidification of the tumoral microenvironment, as well as protein, fatty acid (FA) and ATP synthesis. On the other hand, glutamate (Glu) is taken by cells via glutamate transporter 1 (GLT1) and glutamate/aspartate transporter (GLAST). Also, the Glu is transported outside de cell by the glutamate-cysteine antiporter (XCT) System, in exchange with cysteine. The glutamate is amidated to form glutamine by glutathione synthetase (GS). Furthermore, in the cytosol, glutamate participates in the biosynthesis of GSH and non-essential amino acids. In addition, the Asp derived from Glu is transformed by asparagine synthetase (ASPS) to asparagine (Asn), which inhibits apoptosis. Abbreviation: Proline (Pro), Alanine (Ala), Glucosa (Glu), lactate dehydrogenase (LDH), and nicotinamide adenine dinucleotide (NADPH). ↑ means activation, and inhibition. The figure was created with BioRender.com.

#### 3.1.3. Glutamate Dehydrogenase

The glutamate produced by GLS is metabolized by GDH to αKG and ammonia for TCA cycle anaplerosis [267]. The glutamate oxaloacetate transaminase and glutamate-pyruvate transferase synthesize αKG without releasing ammonia [267,320]. GDH activity is reversible, forming either glutamate or αKG and ammonia, but the αKG production is favored in gliomas, this reaction is activated by mTOR and ADP, deficit energy [286]. SF188 glioblastoma cells, glucose deprivation is accompanied by a large increase in the activity of GDH. The increased GDH activity is suggested to be indirectly regulated by AKT, through its effects on glucose metabolism [321]. αKC can also be synthesized by IDH, mutations in IDH1 are often detected in grade II or III gliomas [322]. Orthotropic grafts with the combination of IDH1 mutated and silenced GDH1,-2 glioma cells demonstrated a significant reduction in the tumor volume compared with grafts of control cells [323]. The oncometabolite D-2 hydroxyglutate (D2HG), is synthesized from mutant IDH1, consuming NADH and to the loss of proper enzymatic activity [324].

### 3.2. Regulation of the Autophagy by Glutamine

In cancer cells, high glutamine consumption is correlated to glycolysis dependency and proliferation. Consequently, glutamine dependency is correlated with autophagy. Autophagy can partially restore cellular glutamine levels by macropinocytosis, recycling intracellular proteins, and extracellular compartments [325]. Studies in glioblastoma cells demonstrated that serine availability supports glioma cell survival following glutamine starvation. Serine synthesis is mediated through autophagy rather than glycolysis [326]. Also, it has been observed that glutamine starvation in glioblastoma cells promotes PGK1 acetylation leading to activation of the VPS34-Beclin1 complex to initiate autophagosomal formation [115]. Furthermore, glutamine starvation induces the inhibition of mTOR, activation of TFEB transcriptional factor, and increases ROS production as well as the FOXO activation, which inhibits mTOR via GLS [327,328,329].

Autophagy activation is correlated with cell survival during nutrient starvation [330]. Research observations indicate that autophagy reduces the survival threshold of several apoptosis-resistant cancer cells upon certain chemotherapies [327]. Tanaka et al. demonstrated that the pharmacological inhibition of mTOR and GLS induces a synergic cell death in vitro and in vivo in glioblastoma models [280]. A phase I clinical trial (NCT03528642) for IDH mut glioma testing with CB-839 (Telaglenastat) a GLS inhibitor showed that Telaglenastat decreases the intracellular Glu and GSH levels in glioma cells and increases the radiotherapy efficacy in vivo [235].

On the other hand, different glutamine metabolites have a distinct effect on autophagy, such as ammonia, α-ketoglutarate, and glutamate. The effect of ammonia on autophagy is dose-dependent and tissue-specific [331]. For instance, high ammonia concentrations change the lysosomal pH and negatively affect autophagic flux [332] whereas α-KG and Leucine are both inhibitors of mTOR [327]. Also, it has been observed that glutaminolysis induces glutathione (GSH) synthesis via glutamate, thus GSH inhibits autophagy by suppressing the ROS/HIF1α/BNIP3 signaling [327]. However, the persistent inhibition of autophagy due to the sustained activation of mTORC1 could lead to drastic mTORC1-dependent cell death “lately called glutamoptosis”. Glutamoptosis is a particular form of apoptotic cell death during unbalanced activation of glutaminolysis during nutritional restriction [333].

### 3.3. Glutaminolisis as a Target of Therapeutic Drugs in Gliomas

Different therapeutic strategies were designed to target glutamine addiction (Table 1), including glutamine analogs (Acivicin and 6-diazo-5-oxo-L-norleucine) [231,232]. The treatment of glioma cells with the glutamine analog 6-diazo-5-oxo-L-norleucine (DON) in combination with L- asparaginase induced more autophagy and apoptosis than single-drug alone. Both compounds deplete asparagine levels by the reduction of exogenous asparagine by L-asparaginase and by the inhibition of asparagine synthetase in the cells, avoiding the proliferation of glioma cells [232]. 

The glutamine depletion by L-asparaginase and compound 968 is another target against glioma-glutamine addiction [231,232,233]. The glutamine antagonis JHU-083 reduced glioma cell growth and disrupted mTOR signaling in glioma cells [239]. The inhibition of GLS by BPTES in glioblastoma cells decreases glutamate, and α-KG levels, and reduces the growth of mutant IDH1 cells compared to those expressing wild-type IDH1 [234]. Other strategies include CB-839 a GLS1 inhibitor, or the inhibition of the kidney-type GLS2 (KGA or GAC siRNA) [235,236,237].

Glutamate uptake induction by GLT1 overexpression inhibits growth proliferation in glioma in vivo and in vitro [240]. The glutamate transport inhibition by the inhibition of the cysteine/glutamate antiporter xCT by sulfasalazine caused a selective, apoptotic, caspase-mediated cell death of GBM cells in vitro and a xenograft model of GBM. While no clinical response to sulfasalazine was observed in clinical trials with terminal patients with glioblastoma [241]. The inhibition of the aspartate glutamate transporter (GLAST), in glioma-bearing mice, significantly increased survival by decreasing GLAST expression and inducing apoptosis [242]. 

The GS inhibition by actinomycin D or 5-azacytidine decreased proliferation in D54-MG human glioblastoma cells [231]. Other compounds such as the natural dipeptide carnosine decreased the expression of GS in a translation level in the glioblastoma cell lines U87 and U251, promoting the degradation of GS through the proteasome pathway, shortening the protein half-life, and reducing its stability [238]. GS overexpression in the C6 glioma cell line, decreased proliferation and migration [310], reversing lactate effects (LDH-A inhibitors) [267,334]. The enhanced aerobic glycolysis of glioblastoma cells supports survival in hypoxic conditions, lactate facilitating invasiveness, and the implementation of anaplerotic conditions for supporting the demands of these proliferative cells. The genetic heterogeneity of gliomas, including surrounding microenvironment and plasticity, influences the glutamine requirements of gliomas in vivo, therefore further information needs to be elucidated for the role of glutamine in different gliomas.

## 4. Conclusions

Metabolic reprogramming is one of the hallmarks of brain cancer. Glioblastoma presents metabolic alterations in the central carbon and energy metabolism, including glycolysis and glutamine metabolism, which promote and enhance the malignancy of glioma cells. Interestingly, metabolites and enzymes that are components of this metabolism promote and enhance glioma cell malignancy and induce the activation of signaling pathways, including tyrosine kinase receptors (TKR)/PI3K/AKT/mTOR, TKR/Ras/RAF/MAPK/ERK and transcriptional factors (HIF-1, p53, and c-myc). Therefore, a broad study of the mechanisms involved in glioma malignancy is necessary to offer a valid and novel anti-cancer therapy.

## Figures and Tables

**Figure 1 ijms-24-17633-f001:**
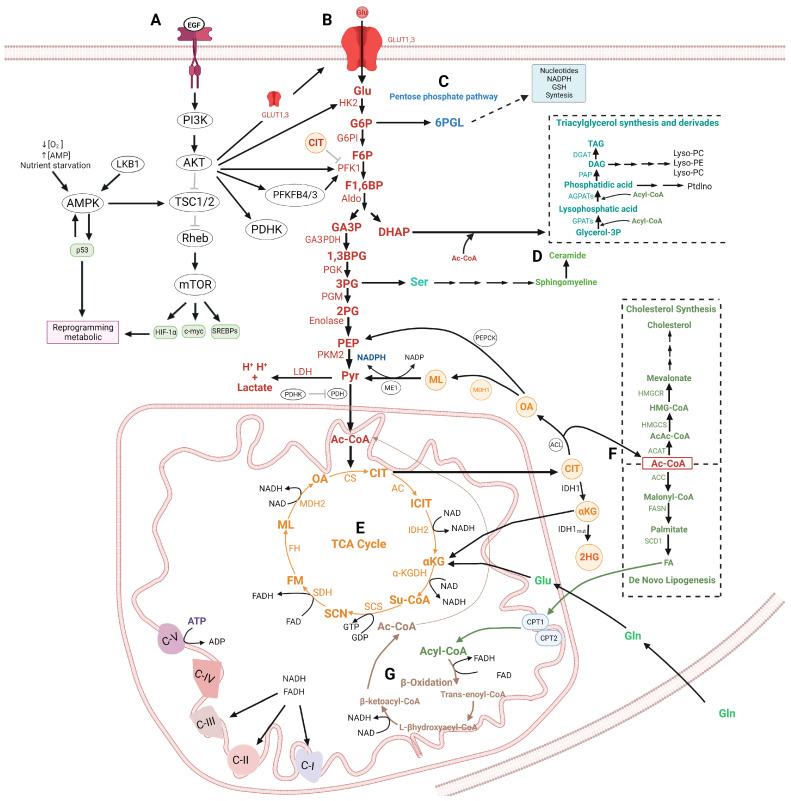
Interconnection of glycolysis with other metabolic pathways. (**A**) Regulation of cellular metabolism by PI3K/AKT/mTOR pathway and AMPK. (**B**) Glycolysis pathway. Hexokinase 2 (HK) phosphorylates at glucose to obtain glucose-6 phosphate (G6P), which is converted to fructose-6-phosphate (F6P) by glucose-6-phosphate isomerase (G6PI). The F6P is converted to fructose-1,6-biphosphate (F1,6BP), subsequently the F1,6BP, F1,6BP are cut by aldolase (Aldo) to produce dihydroxyacetone phosphate (DHAP), and glyceraldehyde-3-phosphate (GA3P). GA3P is transformed by glyceraldehyde-3-phosphate dehydrogenase (GA3PDH) to 1,3-bisphosphoglycerate (1,3BPG), which is converted to 3-phosphoglycerate (3PG) by a phosphoglycerate kinase (PGK). 3PG is converted to 2-phosphoglycerate (2PG) by phosphoglycerate mutase (PGM). Next, 2PG is metabolized to phosphoenolpyruvate (PEP) by enolase (ENO). PEP forms pyruvate (Pyr) by the action of pyruvate kinase (PKM2). Pyr is transformed into Lactate and H+ by Lactate dehydrogenase and acetyl-CoA by the pyruvate dehydrogenase (PDH (PDH). (**C**) The G6P could be redirected to the pentose phosphate pathway (PPP). (**D**) DHAP could be redirected to the triacylglycerol synthesis. 3PG can be a precursor by ceramide synthesis. (**E**) Krebs cycle. The TCA cycle starts with the combination of acetyl-CoA and oxaloacetate (OA) to produce citrate (CIT). CIT is converted into isocitrate (ICIT), which is transformed at α-ketoglutarate (αKG) with the NADH generation, Subsequently, the αKG is converted at succinyl-CoA (Su-CoA) producing NADH. Su-CoA is transformed at succinate (SCN) with the production of GTP. Next, the SCN is converted at fumarate (FM) and releases FADH2. The FM is metabolized to malate (ML), subsequently, the ML is transformed to OA and generates NADH. The OA again reacts with Ac-CoA to continue the cycle. Citrate obtained from the TCA cycle is released at cytosol and converted to Ac-CoA and OA by ATP-citrate lyase (ACLY). (**F**) The Ac-CoA is a substrate by de novo the fatty acids and cholesterol synthesis. (**G**) β-Oxidation. The fatty acids produced by De novo lipogenesis synthesis can be metabolized by the β-Oxidation pathway. Abbreviation: Phosphoinositide 3-kinase (PI3K), protein kinase B (AKT), mammalian target of rapamycin (mTOR), 5′AMP-activated protein kinase (AMPK), hypoxia inducible factor-1 (HIF-1), Sterol Regulatory Element Binding Proteins (SREBP-1), phosphofructo-2-kinase/fructose-2,6-biphosphatase (PFKFB4,-3), pyruvate dehydrogenase kinase1 (PDHK1), phosphofructokinase Type 1 (PFK1), citrate synthase (CS), aconitase (AC), isocitrate dehydrogenase 1, -2 (IDH1,-2), α-ketoglutarate dehydrogenase enzyme complex (α-KGDH), succinyl CoA synthetase (SCS), succinate dehydrogenase (SDH), fumarate hydratase (FH), Malate dehydrogenase (MDH2), Malic Enzyme (ME), ATP-citrate lyase (ACL), acetyl-CoA carboxylases (ACC), acetyl-CoA acetyltransferase (ACAT), fatty acid synthase (FASN), Stearoyl-CoA Desaturase (SCD), Hydroxymethylglutaryl-CoA Reductase (HMGCR), HMGCS: Hydroxymethylglutaryl-CoA Synthase (HMGCS), glycerol-3-phosphate acyltransferases (GPATs), generating lysophosphatidic acid, 1-acylglycerol-3-phosphate acyltransferases (AGPATs), diglyceride acyltransferase (DGAT) phosphatidate phosphatase (PAP), glutamine (Gln), glutamate (Glu). ↑ activation, ⊥ inhibition. The figure was created with BioRender.com.

**Figure 3 ijms-24-17633-f003:**
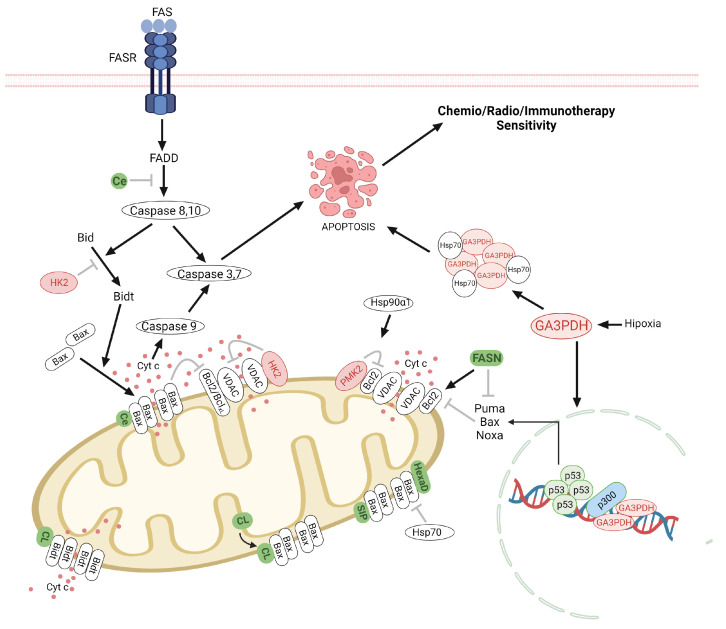
Regulation of apoptotic process by glycolysis and lipid metabolism in glioma. The apoptosis is activated via death receptors (extrinsic) and mitochondrial (intrinsic). The extrinsic pathway is activated by death receptors from the tumor necrosis factor (TNF) family such as Fas (Apo/CD95) and TNF-related apoptosis-inducing ligand (TRAIL) receptors between others, both located on the cell surface, once the Fas/Fas R complex, they recruit at Fas-associated death domain (FADD) protein, which is responsible for recruiting and autoactivation of initiator procaspase-8, which in turn promotes cleaves and the catalytic activation of the effector caspase-3. Furthermore, caspase-8 hydrolyzes at Bid pro-apoptotic protein and generates tBid, which induces the Bax mitochondrial translocation inducing the release of cyt c from mitochondria to cytosol and the subsequent activation of pro-caspase9. On the other hand, an apoptotic pathway also is regulated by Hexokinase2 (HK2), pyruvate kinase (PKM2), glyceraldehyde 3-Phosphate Dehydrogenase (GAPDH), Hexadecenal (HexaD), ceramide (Ce), and sphingosine-1-phosphate (SIP). HK2, also bound voltage-dependent anion channel (VDAC) in the mitochondrial inhibiting apoptosis. The PKM2 inhibits apoptosis through binding and phosphorylation of Bcl-2, whereas Hsp90α1 mediates the binding between Bcl-2 and PKM2. Nuclear GA3PDH induces the activation of p53 and inhibition of Hsp70 at the cytoplasmic level. HexaD, Ce, CL, and SIP are inhibitors of Bax pro-apoptotic proteins. Furthermore, Cl inhibits at Bidt and fatty acid synthase (FASN), is an activator of bcl2 antiapoptotic proteína and an inhibitor of pro-apoptotic proteins including Puma, Noxa, and Bax, which are positively regulated by p53. Abbreviation: B cell lymphoma 2 (Bcl-2), B cell lymphoma-extra-large (Bcl-xL), Fas-associated via death domain (FADD), caspases (Casp)-8,-10,-9 and -3 ↑ activation, ⊥ inhibition. The figure was created with BioRender.com.

**Figure 4 ijms-24-17633-f004:**
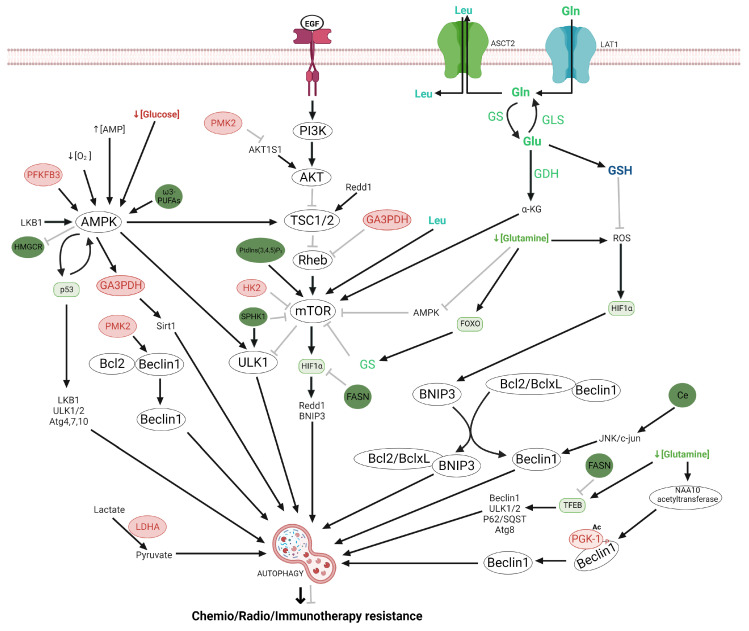
Regulation of autophagy by glycolysis, lipid metabolism, and glutamine in glioma. Autophagy could be regulated by Hexokinase2 (HK2), which directly binds and inhibits the mammalian target of rapamycin complex1 (mTORC1) leading to autophagy. Phosphofructo-2-kinase/fructose-2,6-biphosphatase3 (PFKFB3) induces autophagy by phosphorylating at 5′AMP-activated protein kinase (AMPK); glyceraldehyde 3-Phosphate dehydrogenase (GA3PDH) inhibits mTOR by binding to Rheb. PGK1 activates Beclin1 and induces autophagy; pyruvate kinase (PMK2) activates autophagy by inhibiting phosphorylation at AKT1 substrate 1 (AKT1S1) a mTORC1 inhibitor. Fatty acid synthase (FASN) inhibits hypoxia-inducible factor-1 α (HIF-1α) and transcription factor EB (TFEB), Ceramide (Ce) induces autophagy by activating at c-Jun N-terminal kinase (JNK)/c-Jun pathway. Omega-3-polyunsaturated fatty acids (ω3-PUFAS) activate at AMPK; Phosphoinositides (3,4,5)triphosphate (PtdIns (3,4,5) P3) activate at mTOR. Hydroxymethylglutaryl-coenzyme A reductase (HMGCR) activates at AMPK. Abbreviations, leucine (leu)Glutamine synthetase, glutamine (Gln), glutamate (Glu), Glutaminase (GLS), Glutamate dehydrogenase (GDH), Glutamine transport via alanine/serine/cysteine transporter 2 (ASCT2), glutathione (GSH), Forkhead box O (FOXO). ↑ activation, ⊥ inhibition. The figure was created with BioRender.com.

**Table 1 ijms-24-17633-t001:** Treatment options by the inhibition of glycolysis and gltaminolysis.

Target	Treatment	Clinical and Preclinical Trials
**Glycolysis**		
Glycolysis pathway	Methylene blue	Increased OCR and decreased ECAR and lactate production in glioblastoma cell lines [223].
Enolase	HEX and POMHEX	Decrease in TCA cycle metabolites using a glioma cell expressing ENO2 but no ENO1 [136].
Pyruvate Dehydrogenase Kinases (PDHK)	Dichloroacetate (DCA)	Increased the oxidative phosphorylation in vitro. Also, induces cell death in glioma cells, through the generation of ROS with a decrease in p53, HIF-1, p21, proliferating cell nuclear antigen (PCNA), hexokinase 2, VEGF, and α-ketoglutarate [224,225].
6-phosphofructo-2-kinase/fructose-2,6-bisphosphatase 3 (PFKFB3).	3-(3-Pyridinyl)-1-(4-pyridinyl)-2-propen-1-one (3PO)	In complement to bevacizumab, decreased cell proliferation and increased apoptosis in vitro, whereas delayed tumor growth and improved survival using in vivo [226]
Syk inhibitor	R406	Shifts the glioma cells from a glycolytic profile toward OXPHOS metabolism, and increases apoptosis [227].
Monocarboxylate transporter 4 (MCT4)	Acriflavine	Decrease cell proliferation and tumoral growth [228,229].
Histone deacetylase (HDAC)	Vorinostat	Decrease intracellular lactate, cell viability, and tumor progression [230].
**Glutaminolysis**		
Glutaminase (GLS)	Compound 968	Inhibit the glioma cell growth with the Notch pathway blocked [231,232,233]
	bis-2-(5-phenylacetamido-1,2,4-thiadiazol-2-yl) ethyl sulfide (BPTES).	Slows GBM growth cell with IDH1mut [234].
	CB-839	Reduced proliferation, angiogenesis, and modestly increased apoptosis in glioma stem-like cells with IDH1mut [235,236,237].
Glutamine Synthetase (GS)	Actinomycin D, 5-azacytidine.	Decreased proliferation in glioma cells [231].
	Carnosine	Suppressed growth and migration in glioma cells [238].
Glutamine	Acivicin, 6-diazo-5-oxo-L-norleucine (DON).	Inhibit the growth of glioma cells [231].
	6-diazo-5-oxo-L-norleucine (DON) plus L-asparaginase.	Induces apoptosis and autophagy in glioma cells by the depletion of asparagine via inhibition of asparagine synthetase [232].
	JHU-083	Inhibits cell growth and mTOR signaling [239].
Glutamate	GLT1 cDNA overexpression.	Inhibits growth proliferation in glioma cells [240].
xCT	Sulfasalazine	Induce apoptosis in GBM cells [241].
GLAST	shGLAST	Increases survival in glioma-bearing mice [242].

## Data Availability

No new data were created or analyzed in this study. Data sharing does not apply to this article.

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
