# Peer review of "Role of Glycolytic and Glutamine Metabolism Reprogramming on the Proliferation, Invasion, and Apoptosis Resistance through Modulation of Signaling Pathways in Glioblastoma"

_ijms, 2023, doi:10.3390/ijms242417633_

Round 1
Reviewer 1 Report
Comments and Suggestions for Authors
The manuscript by Trejo-Solís et al presents an interesting and very exhaustive review on metabolic aspects in glioblastoma multiforme. The manuscript is also very well written and well organized. The main problem in my opinion is that many of the paragraphs presented are structured in a way that is too long in the sense that some concepts, for example such as glycolysis (chapter 29) or glutaminase pathway (chapter 5) are known to most readers. Consequently, these paragraphs should be reduced and simplified to make the manuscript more fluent.
Minor comment
Page 4 line201 10.1158/1541-7786.MCR-201 05-0061 should be converted in the appropriate reference
Author Response
Referee #1:
Comment to the Authors:
The manuscript by Trejo-Solís et al presents an interesting and very exhaustive review on metabolic aspects in glioblastoma multiforme. The manuscript is also very well written and well organized. The main problem in my opinion is that many of the paragraphs presented are structured in a way that is too long in the sense that some concepts, for example such as glycolysis (chapter 29) or glutaminase pathway (chapter 5) are known to most readers. Consequently, these paragraphs should be reduced and simplified to make the manuscript more fluent.
Response to major comments:
Thank you for your observations. To get a more fluent text, we exclude redundant information of chapter 2 related to glycolysis (page 4, line 118 to 144) and chapter 5 related to glutaminase pathway (page 25, line 1158 to 1184) as suggested.
Minor comment
Comment #1
Page 4 line201 10.1158/1541-7786.MCR-201 05-0061 should be converted in the appropriate reference.
Response #1.
Thank you for the observation, the reference has been properly inserted.
Reviewer 2 Report
Comments and Suggestions for Authors
The respected authors provide a very well-organized review article focusing the impact of the glycolytic pathway and glutamine pathway in glioblastoma resistance to the apoptotic pathway. The review needs the following major considerations that should be done to improve the quality of the work:
1- The general information of the glycolytic pathway should be removed from the text, and they all could be explained in an informative scheme summarizing this pathway.
2- The link of the glycolytic pathway with TCA in glioma should be well described, and then discuss its impact in regulation of apoptosis resistance.
3- The link between lipid metabolism and glycolytic needs briefly explained and then discussed on the context of glioblastoma apoptosis resistance. All metabolic pathways are connected and cannot be discussed without each others.
4- The impact of the autophagy/glycolytic pathway and autophagy/glutamine pathway needs to be described in a separate section and how it affects the apoptosis resistance. It is scattered right now in different places.
5- A section related to human studies describes these relations should be added any clinical tiral which are active on this pathway needs to be considered.
6- The impact of metformin as a major regulator of glucose metabolic pathway needs to be explained in a different section.
Minor CHanges:
1- The review will benefit from adding more schematics
2- The review will benefit from a graphical abstract which provides the take home message.
Author Response
Referee #2:
Comment to the Authors:
The respected authors provide a very well-organized review article focusing the impact of the glycolytic pathway and glutamine pathway in glioblastoma resistance to the apoptotic pathway. The review needs the following major considerations that should be done to improve the quality of the work:
Comment #1
1- The general information of the glycolytic pathway should be removed from the text, and they all could be explained in an informative scheme summarizing this pathway.
Response #1.
Thank you for your observations. To get a more fluent text, we exclude redundant information of chapter 2 related to glycolysis (page 4, line 118 to 144). However, the glycolytic pathway is already integrated in the Figure 1.
Comment #2
2- The link of the glycolytic pathway with TCA in glioma should be well described, and then discuss its impact in regulation of apoptosis resistance.
Response #2
Thank you for your comment, we integrate a new figure (Figure 1) to explain this point and the following text to the manuscript as follows (page 4, line 164):
A link between the glycolytic pathway and TCA has been established with citrate as an intermediary and documented previously (PMID 37717858, 34205414, 2791284). Citrate is an inhibitor of PFK1, a key regulator of glycolysis, and PFK2, which produces F2,6-BP, an activator of PFK1. Then, the interaction between both pathways is well established. Moreover, citrate as metabolite is linked to the grade of aggressiveness in cancer through the apoptotic process. It has been reported that citrate inhibits the expression of BCLxL and MCL1, both members of the anti-apoptotic BCL2 family. Besides, citrate induces the expression of pro-apoptotic proteins such as Bax, caspase 3, and 9. Then, the regulation of both pathways is a key factor in the resistance of apoptosis (PMID 37717858). Furthermore, glycolytic intermediates are precursors for the synthesis of acyl triglycerides, cholesterol, and fatty acids (Figure 1).
Comment #3
3- The link between lipid metabolism and glycolytic needs briefly explained and then discussed on the context of glioblastoma apoptosis resistance. All metabolic pathways are connected and cannot be discussed without each others.
Response #3
Thank you for your comment, we integrate this point to the manuscript as follows (page 17, line 821):
Therapeutic implications of glycolysis and lipid metabolism in the apoptosis regulation in glioma
Glycolysis is related to other metabolic pathways, including lipid metabolism as we briefly describe in Figure 1. Besides, it has been proposed that resistance to cell death in glioma cells is due to the dysregulation of apoptotic and autophagic pathways through the overexpression for AMPK, TKR/PI3K/AKT/mTOR, p53, and HIF-1 signaling pathway (PMID 30486451). AKT can modulate both processes by inducing glycolysis and lipids metabolism, which supply the energetic and cellular components for proliferation and cellular survival under several stress conditions (PMID 37190124).
On the other hand, it has been described that the glycolytic pathway modulates apoptosis via PMK2, GADPH, and HK2 (PMID 33380833) (Figure 3), whereas HK2 and PMK2 inhibit the apoptosis in cancer cells to confer cellular protection under starvation conditions such as glucose and glutamine promoting the migration and high resistance to therapies (PMID 25176644, 28915575).
Dong-Qiang et al demonstrated that 3-Bromopyruvate (3BPr) induces apoptosis in CD133+ U87 glioma cells by increasing the Bax levels and the caspase3 activity (PMID 26453119). 3-BPr recruits pro-apoptotic BAX protein to mitochondria promoting the inhibition of Bcl-2, and Mcl-1 and the caspase-3 activity (PMID: 15695406, 28123308, 22298255). 3-BP blocks the binds of HK2 to the mitochondrial membrane and in turn, promotes the release of mitochondrial AIF to cytosol inducing cell death (PMID 22298255). It has been suggested that the combination of 3-BPA with chemotherapeutic drugs can increase the chemoresistance in cancer cells by depleting the ATP, lipids, protein, nucleotides levels, and the DNA repair systems (PMID 19383331). The administration of 3-BPA has been also evaluated in stage IV metastatic melanoma and patients with advanced hepatocellular carcinoma (PMID 36230492, 22328020). Lonidamine, another inhibitor of HK2 was administered in clinical trials for the treatment of several types of cancer including ovarian, lung, and breast cancer (PMID 12730701, 9336357, 10426128, 12377958). However, the hepatic and pancreatic toxicities presented by lonidamine has limited its clinical success (DOI: 10.1007/s002800050460).
Also, the lipids play an important role in the regulation of apoptosis in both the intrinsic and extrinsic pathways (Figure 3). In the intrinsic pathway the regulation of apoptosis is carried out by mitochondrial lipids such as cardiolipin (CL), ceramide, and sphingosine-1-phosphate (PMID 24007978, doi: 10.1016/0076-6879(87)48046-4). Cardiolipin induces the formation of BAX and BAK1 oligomers, for the formation of pores and consequently the release of cytochrome C into the cytoplasm, which induces the activation of caspases (PMID 18171672). Cardiolipin has been reported to interact with tBid by translocating to the mitochondrial outer membrane, thereby inducing the release of cytochrome C into the cytosol. Sorice et al reported that glioma cells stimulated with Fas induced the activation of caspase-3, the formation of CL/Bid complex (PMID 15181455). Mitochondrial ceramide (Ce) is a lipid that is produced upon pro-apoptotic stimuli and represents the main requirement for BAX oligomerization, thus forming ceramide channels (PMID 16167171). Glioma cells treated with indomethacin presented apoptosis with an increase in the ceramide generation and protein phosphatase A2 (PPA2), AKT inactivation, translocation of Bax from the cytosol to mitochondria,caspase3 activation and down-regulation of Mcl1. The apoptotic events were also duplicated with the administration of C2-ceramide andLY294002 (Akt inhibitor) (PMID 29470962). It has been demonstrated that TMZ and radiation induce the sphingomyelase activity, which hydrolyzes sphingomyelins to ceramides inducing apoptosis (PMID 32977496). However, cells can transform ceramides sphingosine-1-phosphate to evade apoptosis (PMID 32977496).
On the other hand, sphingosine-1-phosphate (S1P) upon degradation by sphingosine-1-phosphate lyase 1 forms hexadecenal, which interacts with BAK1 and BAX respectively, inducing oligomerization and cytochrome C release (PMID 22385963). Amaegberi et al demonstrated that 2-hexadecenal (HexaD) inhibits cell proliferation and induces apoptosis in glioma cells (PMID 31038222). Cholesterol also plays a role in apoptosis, since the presence of cholesterol inhibits Bax and contributes to the decreased cellular ability to induce mitochondrial outer membrane permeabilization. In addition, cholesterol can inhibit the insertion and activation of Bax, thus preventing apoptosis (PMID 18084240, 18593925). Studies have reported the possible role of polyunsaturated fatty acids (PUFA) in cancer cell apoptosis through Bcl2, cytochrome C, and caspases (PMID 26528354, 18361731). Oleic acid is an anticancer agent that induces autophagy and apoptosis. The role of stearic acid as an inhibitor of invasion, proliferation, and inducer of apoptosis in cancer cells has also been studied, however, the mechanisms remain unclear (PMID 28900281). Conjugates of temozolomide with fatty acids such as palmitic acid, linoleic acid, and oleic acid increase the enhancer TMZ efficacy, overcoming the resistance at TMZ induced by MGMT glioma models in vitro and in vivo (PMID 37182806). Furthermore, it has been demonstrated that enzymes involved in the lipid metabolism regulate the apoptosis. FASN inhibits apoptosis by suppressing the Puma, Bax, and Noxa pro-apoptotic proteins (PMID 37768124). FASN inhibitors induces apoptosis in glioma cells by activation of caspases 3 and downregulation of Bcl-2 protein (PMID 16969344). These studies suggest that drug sensitivity in glioma cells could be enhanced through of inhibitors from glycolysis and lipid metabolism in combination with an activator of the apoptosis pathway.
Comment #4
4- The impact of the autophagy/glycolytic pathway and autophagy/glutamine pathway needs to be described in a separate section and how it affects the apoptosis resistance. It is scattered right now in different places.
Response #3
Thank you for your comment, we added the following text to the manuscript (page 19, line 940):
Regulation of the autophagy by glycolysis and lipid metabolism
Autophagy is a catabolic process that can promote cell death or survival in cancer cells and the relevance of autophagy in glioma has been extensively revised previously (PMID 32707662). It has been reported that the enzymes and metabolites related to glycolysis participate in the regulation of autophagy (PMID 33380833, 37190124) (Figure 4).
Autophagy is a highly conserved and regulated process to digest organelles and proteins and recycle cytoplasmatic components, which can protect the cancer cells from harsh environment increasing their capacity for adaptation and survival (PMID 18191218). In cancer cells, autophagy can increase or decrease with chemotherapy agents playing an essential role in cell death (suppressor tumoral) or cell proliferation (promotor tumoral) depending on the cellular context. Therefore, the modulation of the autophagy is an attractive target for anti-tumor therapies.
Glioma cells present high rates of drug resistance, and autophagy contribute to the resistance to treatment. TMZ is generally used as therapy for the first line of GBM, however, TMZ resistance is modulated by autophagy (PMID 23383259). Also, it has been observed that bevacizumab (VEGFR inhibitor) also shows low efficacy on glioma cells due to the promotion of autophagy in hypoxic conditions (PMID: 22447568). Under hypoxia, HIF1 increases the Bcl-2 interacting protein 3 (BNIP3), which is released at Beclin 1 from Becli1/Bcl-2 complex inducing autophagy in U87 and T96G) cells. Bevacizumab plus chloroquine increases the glioma cell response to treatment (PMID 22447568).
Autophagy is generally induced due to a diminution in the ATP levels, nutrient deprivation, and hypoxia inducing the AMPK activation and in turn the inhibition of mTOR (PMID 37190124). Glucose deprivation induces autophagy through HK2, which binds at mTOR inhibiting its activity (PMID 26075878). mTOR inhibits autophagy by negatively regulating the activation of unc-51-like autophagy activating kinase 1 (ULK1) complex, which is necessary for the phagophore formation (PMID 26075878). Then, it has been suggested that a HK2 molecular link between glycolysis and autophagy ensures cellular energy supply (PMID 26075878). The impact of metformin as a major regulator of glucose metabolic pathway has been exten-sively review elsewhere (PMID 31952173). Moreover, Sesen et al demonstrated that metformin (HK2 inhibitor) promotes apoptosis and autophagy by increases at LC3B and Beclin 1 and decreases sequestosome 1 (SQSTM1) levels in glioma cells by activating Redd1 (TSC2 activator) and AMPK and inhibiting mTOR, S6K and 4EBP1 increasing the cytotoxic effects of radiotherapy and temozolomide (PMID: 25867026, 24240433). A combination of arsenic trioxide plus metformin promoted autophagy and apoptosis in glioma cells through the activation of AMPK/FOXO3 (PMID 23197693). FOXO3 regulates the transcription of autophagic genes such as ULK1/2, Beclin1, Atg4,-5,-8 and-14 (PMID 31952173, 30486451, 32707662).
Nuclear PFKFB3 induces autophagy by phosphorylating at AMPK. On the other hand, the suppression of PFKFB3 by 3-(3-pyridinyl)-1-(4-pyridinyl)- 2-propen-1-one in HCT-116 cells induces autophagy as a pro-survival mechanism through the ROS generation, decrease in glucose levels and inactivation of p70S6K and S6 (PMID 24451478). Blum et al demonstrated that pharmacological inhibition of Ras in glioma cells caused a reduction in HIF-1α expression via inhibition of the PI3K/AKT pathway and in turn decreases of PFKFB3 and glycolysis, promoting the cell death (PMID: 15705901). On the other hand, it has been reported that under glucose starvation AMPK phosphorylated at GA3PDH inducing its nuclear translocation, whereas to bind at Sirt1 histone deacetylases inducing a rapid initiation of autophagy (PMID 26626483). Sirt1 can bind and activate to Atg5, atg7, and atg8 autophagic complex (PMID 18296641). Also, it has been reported that GA3PDH inhibits mTOR by binding at Rheb (PMID 19451232). Dando et al reported that the inhibition of mitochondrial uncoupling protein 2 (UCP2) protein induces the ROS generation and in turn nuclear translocation GA3PDH, inactivation of Akt/mTOR/P70S6K signaling pathway promoting the apoptosis independent of caspases and autophagy mediated by Beclin1 in pancreatic adenocarcinoma cells (PMID 28962872). PGK1 another glycolytic enzyme also modulates autophagy. Hypoxia or glutamine starvation in glioma cells promotes the phosphorylation of S301 Beclin1 by PGK1 through the mTOR inhibition which promotes the activation from autophagic initiation complex, which contributes to the brain tumoral aggressiveness (PMID 28238651). PGK1 promotes radioresistance and invasion in glioma cells.
In addition, several studies demonstrated that PKM2 modulates autophagy. It has been reported that PMK2 activates autophagy by inhibiting phosphorylation at AKT1 substrate 1 (AKT1S1) a mTORC1 inhibitor, which facilitates mTOR activation (PMID 26876154). It has been reported a positive loop between PKM2 and mTORC1, that mTORC1 increases PKM2 expression and in consequence induces an accumulation and utilization of metabolic intermediates (PMID 26876154). PKM2 knockdown lung cancer cell lines increase the radiosensitivity at ionizing radiation by promoting apoptosis and autophagy (PMID 25444918). PKM2 also induces survival mechanism autophagy by the phosphorylation at Thr119 Beclin-1 (PMID 30906218). pThr119 Beclin-1 is release from Bcl2/Beclin1 complex for induces autophagy. On the other hand, it has been proposed that Beclin1 could show a pro-apoptotic effect by preventing of function of Bcl-2 and BclxL in glioma cells (PMID 24535641). The PMK2 overexpression by histone methyltransferase G9a inhibition in glioma cells modulates the LC-3II and YAP-1 levels promoting autophagy (PMID 33380833). Another study demonstrated that LDHB promotes autophagy by contributing to lysosomal acidification and formation of the autophagosome and thus cancer cell survival (PMID 30443978).
Also, autophagy could be modulated by several lipid species such as sterols, sphingolipids, and phospholipids derived from glycolysis intermediaries (Figure 4). These lipids and lipid-related processes act on signaling processes throughout all stages involved in autophagy such as initiation, autophagosome development, and autolysosome maturation. Phosphoinositides act as key modulators of autophagy (PMID 37768124). PtdIns(3,4,5)P3 participated in the regulation of autophagy by activation of mTOR signaling pathway (PMID 37768124). The phagophore is formed from PtdIns3P-enriched sites, that allow the recruitment of diverse PtdIns3P- binding proteins (PMID 23305670), PtdIns5P synthesis is required to autophagosome formation (PMID 25578879), PtdIns4P reduces the autophagic flux by blocking lysosome and autophagosome fusion (PMID 21703229).
Sphingolipids are other molecules that participate in the induction of autophagy (PMID 21703229), one of them is the ceramide. Due to increased ceramide levels, JUN/c-JUN promotes autophagy through MAP1LC3 and Beclin transcription (PMID 21943220, 19060920). Also, it has been demonstrated that fatty acids regulate the autophagy. In this sense, omega-3-polyunsaturated fatty acids (ω3-PUFAs), such as docosahexaenoic acid (DHA), have been reported to increase autophagy-like cell death mechanism by activating AMPK and dephosphorylating AKT and mTOR in glioma cells (PMID 29192322).
On the other hand, Fatty Acid Synthase (FASN) FASN inhibits the autophagy through mTOR (PMID 33742137). mTOR inactivates by phosphorylate at TFEB promoting its cytoplasmatic localization (PMID 33742137). FASN is overexpressed in human glioma samples and its inhibition by Orlistat induces autophagy in glioma (PMID 24789255). FASN inhibitors, also reduce significantly the tumor volume and angiogenesis by down-regulation of VEGF and HIF-1α (PMID 27601165). Simvastatin a hydroxymethylglutaryl-coenzyme A (HMG-CoA) inhibitor promotes cell death in C6 and U251 glioma by inducing the AMPK, LC3-II Beclin1 activation, inactivation of AKT and in turn mmTOR/ p70 S6 kinase1, also down the mevalonate levels (PMID 21871960). On the other hand, Sphingosine-1-Kinase (SPHK1) modulates the autophagy, then regulates the growth, invasion, and therapy resistance, cancer development, growth, and metastasis (PMID 28360148). SPHK1 induces autophagy through of inhibition of mTOR, up-regulation of ULK1 and Atg proteins in cancer cells (doi:10.3390/cancers15082195).
Then, to increase the drug sensitivity on glioma cells could be necessary therapy that involves inhibitors of the cell metabolism in combination with activators or suppressors of the autophagy pathway.
Thank you for your comment, we integrate this point to the manuscript as follows (page 29, line 1385):
Regulation of the autophagy by glutamine
In cancer cells, high glutamine consumption is correlated to glycolysis dependency and proliferation. Consequently, glutamine dependency is correlated with autophagy. Autophagy can partially restore cellular glutamine levels by macropinocytosis, recycling intracellular proteins and extracellular compartments (PMID 31257175). Studies in glioblastoma cells demonstrated that serine availability supports glioma cell survival following glutamine starvation. Serine synthesis is mediated through autophagy rather than glycolysis (PMID 33468252). Also, it has been observed that glutamine starvation in glioblastoma cells promotes PGK1 acetylation leading to activation of the VPS34-Beclin1 complex to initiate autophagosomal formation (PMID 28238651). Furthermore, the glutamine starvation induces the inhibition of mTOR, activation of TFEB transcriptional factor, and increases in the ROS production as well as the FOXO activation, which inhibits mTOR via GLS (PMID 34973953, 22820375, 22996802).
Autophagy activation is correlated with cell survival during nutrient starvation (PMID 33718649). Research observations indicate that autophagy reduces the survival threshold of several apoptosis-resistant cancer cells upon certain chemotherapies (PMID 34973953). Tanaka et al demonstrated that the pharmacological inhibition of mTOR and GLS induces a synergic cell death in vitro and in vivo in glioblastoma models (PMID 25798620). A phase I clinical trial (NCT03528642) for IDH mut glioma testing with CB-839 (Telaglenastat) a GLS inhibitor showed that Telaglenastat decreases the intracellular Glu and GSH levels in glioma cells and increases the radiotherapy efficacy in vivo (PMID 30220459).
On the other hand, different glutamine metabolites have a distinct effect on autophagy, such as ammonia, a-ketoglutarate, and glutamate. The effect of ammonia on autophagy is dose-dependent and tissue-specific (PMID 33518108). For instance, high ammonia concentrations change the lysosomal pH and negatively affect autophagic flux (PMID 31648987) whereas α-KG and Leucine are both inhibitors of mTOR (PMID 34973953). Also, it has been observed that glutaminolysis induces glutathione (GSH) synthesis via glutamate, thus GSH inhibits autophagy by suppressing the ROS/HIF1α/BNIP3 signaling (PMID 34973953). However, the persistent inhibition of autophagy due to the sustained activation of mTORC1 could lead to drastic mTORC1-dependent cell death “lately called glutamoptosis”. Glutamoptosis is a particular form of apoptotic cell death during unbalanced activation of glutaminolysis during nutritional restriction (PMID 28296535).
Comment #5
5- A section related to human studies describes these relations should be added any clinical tiral which are active on this pathway needs to be considered.
Response #5
Thank you for your comment, examples of human trials have been added in section 2.3 and the table 1.
Comment #6
6- The impact of metformin as a major regulator of glucose metabolic pathway needs to be explained in a different section.
Response #6
Thank you for your comment, this issue has been extensively described by Mazurek.
Then, we integrate this point to the manuscript as follows (page 19, line 965):
The impact of metformin as a major regulator of glucose metabolic pathway has been extensively review elsewhere (PMID 31952173). Moreover, Sesen et al demonstrated that metformin (HK2 inhibitor) promotes apoptosis and autophagy by increases at LC3B and Beclin 1 and decreases sequestosome 1 (SQSTM1) levels in glioma cells by activating Redd1 (TSC2 activator) and AMPK and inhibiting mTOR, S6K and 4EBP1 increasing the cytotoxic effects of radiotherapy and temozolomide (PMID: 25867026, 24240433).
Minor Changes:
Comment #1
1- The review will benefit from adding more schematics
Response #1
Thank you for your suggestion, we added figures 2, 3 and 4 in the manuscript.
Comment #2
2- The review will benefit from a graphical abstract which provides the take home message.
Response #2
The graphical abstract has been added.
Reviewer 3 Report
Comments and Suggestions for Authors
The authors that provided the very good and useful information for reader from the glycolytic and glutamine metabolic network of the cellular pathophysiology of glioblastoma. These informatics may indicate as a therapeutic target in glioma cells. Some minor comments as following:
1. In the figure 1 and 2 have mention the autophagy process, I prefer to see addressing more on this.
2. In figure 1 or 2 should try to combine the some useful drugs on or inhibitor on the signaling pathways.
3. Some background such as cell death (apoptosis) should be enhanced.
Comments on the Quality of English LanguageMinor editing of English language required.
Author Response
Referee #3:
Comment to the Authors:
The authors that provided the very good and useful information for reader from the glycolytic and glutamine metabolic network of the cellular pathophysiology of glioblastoma. These informatics may indicate as a therapeutic target in glioma cells. Some minor comments as following:
Comment 1:
In the figure 1 and 2 have mention the autophagy process, I prefer to see addressing more on this.
Response #1:
Thank you for your comment, we added the following text to the manuscript (page 19, line 940):
Autophagy is a catabolic process that can promote cell death or survival in cancer cells and the relevance of autophagy in glioma has been extensively revised previously (PMID 32707662). It has been reported that the enzymes and metabolites related to glycolysis participate in the regulation of autophagy (PMID 33380833, 37190124) (Fig. ).
Autophagy is a highly conserved and regulated process to digest organelles and proteins and recycle cytoplasmatic components, which can protect the cancer cells from harsh environment increasing their capacity for adaptation and survival (PMID 18191218). In cancer cells, autophagy can increase or decrease with chemotherapy agents playing an essential role in cell death (suppressor tumoral) or cell proliferation (promotor tumoral) depending on the cellular context. Therefore, the modulation of the autophagy is an attractive target for anti-tumor therapies.
Glioma cells present high rates of drug resistance, and autophagy contribute to the resistance to treatment. TMZ is generally used as therapy for the first line of GBM, however, TMZ resistance is modulated by autophagy (PMID 23383259). Also, it has been observed that bevacizumab (VEGFR inhibitor) also shows low efficacy on glioma cells due to the promotion of autophagy in hypoxic conditions (PMID: 22447568). Under hypoxia, HIF1 increases the Bcl-2 interacting protein 3 (BNIP3), which is released at Beclin 1 from Becli1/Bcl-2 complex inducing autophagy in U87 and T96G) cells. Bevacizumab plus chloroquine increases the glioma cell response to treatment (PMID 22447568).
Autophagy is generally induced due to a diminution in the ATP levels, nutrient deprivation, and hypoxia inducing the AMPK activation and in turn the inhibition of mTOR (PMID 37190124). Glucose deprivation induces autophagy through HK2, which binds at mTOR inhibiting its activity (PMID 26075878). mTOR inhibits autophagy by negatively regulating the activation of unc-51-like autophagy activating kinase 1 (ULK1) complex, which is necessary for the phagophore formation (PMID 26075878). Then, it has been suggested that a HK2 molecular link between glycolysis and autophagy ensures cellular energy supply (PMID 26075878). The impact of metformin as a major regulator of glucose metabolic pathway has been exten-sively review elsewhere (PMID 31952173). Moreover, Sesen et al demonstrated that metformin (HK2 inhibitor) promotes apoptosis and autophagy by increases at LC3B and Beclin 1 and decreases sequestosome 1 (SQSTM1) levels in glioma cells by activating Redd1 (TSC2 activator) and AMPK and inhibiting mTOR, S6K and 4EBP1 increasing the cytotoxic effects of radiotherapy and temozolomide (PMID: 25867026, 24240433). A combination of arsenic trioxide plus metformin promoted autophagy and apoptosis in glioma cells through the activation of AMPK/FOXO3 (PMID 23197693). FOXO3 regulates the transcription of autophagic genes such as ULK1/2, Beclin1, Atg4,-5,-8 and-14 (PMID 31952173, 30486451, 32707662).
Nuclear PFKFB3 induces autophagy by phosphorylating at AMPK. On the other hand, the suppression of PFKFB3 by 3-(3-pyridinyl)-1-(4-pyridinyl)- 2-propen-1-one in HCT-116 cells induces autophagy as a pro-survival mechanism through the ROS generation, decrease in glucose levels and inactivation of p70S6K and S6 (PMID 24451478). Blum et al demonstrated that pharmacological inhibition of Ras in glioma cells caused a reduction in HIF-1α expression via inhibition of the PI3K/AKT pathway and in turn decreases of PFKFB3 and glycolysis, promoting the cell death (PMID: 15705901). On the other hand, it has been reported that under glucose starvation AMPK phosphorylated at GA3PDH inducing its nuclear translocation, whereas to bind at Sirt1 histone deacetylases inducing a rapid initiation of autophagy (PMID 26626483). Sirt1 can bind and activate to Atg5, atg7, and atg8 autophagic complex (PMID 18296641). Also, it has been reported that GA3PDH inhibits mTOR by binding at Rheb (PMID 19451232). Dando et al reported that the inhibition of mitochondrial uncoupling protein 2 (UCP2) protein induces the ROS generation and in turn nuclear translocation GA3PDH, inactivation of Akt/mTOR/P70S6K signaling pathway promoting the apoptosis independent of caspases and autophagy mediated by Beclin1 in pancreatic adenocarcinoma cells (PMID 28962872). PGK1 another glycolytic enzyme also modulates autophagy. Hypoxia or glutamine starvation in glioma cells promotes the phosphorylation of S301 Beclin1 by PGK1 through the mTOR inhibition which promotes the activation from autophagic initiation complex, which contributes to the brain tumoral aggressiveness (PMID 28238651). PGK1 promotes radioresistance and invasion in glioma cells.
In addition, several studies demonstrated that PKM2 modulates autophagy. It has been reported that PMK2 activates autophagy by inhibiting phosphorylation at AKT1 substrate 1 (AKT1S1) a mTORC1 inhibitor, which facilitates mTOR activation (PMID 26876154). It has been reported a positive loop between PKM2 and mTORC1, that mTORC1 increases PKM2 expression and in consequence induces an accumulation and utilization of metabolic intermediates (PMID 26876154). PKM2 knockdown lung cancer cell lines increase the radiosensitivity at ionizing radiation by promoting apoptosis and autophagy (PMID 25444918). PKM2 also induces survival mechanism autophagy by the phosphorylation at Thr119 Beclin-1 (PMID 30906218). pThr119 Beclin-1 is release from Bcl2/Beclin1 complex for induces autophagy. On the other hand, it has been proposed that Beclin1 could show a pro-apoptotic effect by preventing of function of Bcl-2 and BclxL in glioma cells (PMID 24535641). The PMK2 overexpression by histone methyltransferase G9a inhibition in glioma cells modulates the LC-3II and YAP-1 levels promoting autophagy (PMID 33380833). Another study demonstrated that LDHB promotes autophagy by contributing to lysosomal acidification and formation of the autophagosome and thus cancer cell survival (PMID 30443978).
Comment 2:
In figure 1 or 2 should try to combine the some useful drugs on or inhibitor on the signaling pathways.
Response #2
Thank you for your comment, Figures 1 and 2 have been modified.
Comment 3:
Some background such as cell death (apoptosis) should be enhanced.
Response #3:
Thank you for your comment, we added the following text to the manuscript (page 7, line 325)
Apoptosis is a cell death mechanism that allows the elimination of damaged cells. In cancer, including glioma, the dysregulation of apoptosis is related to aggressiveness. This process has been extensively reviewed elsewhere and we pointed out the relation with metabolism when is properly (PMID 29259986).
Also, a section related to Therapeutic implications of glycolysis and lipid metabolism in the apoptosis regulation in glioma was added (page 17, line 821):
Round 2
Reviewer 2 Report
Comments and Suggestions for Authors
I have reviewed the revised version of the manuscript. The respected authors carefully followed the comments, and I enjoyed reading a well-organized, right-to-the-point manuscript.